# Polyphenolic Compounds from Andean Berry (*Vaccinium meridionale* Swartz) and Derived Functional Benefits: A Systematic and Updated Review

**DOI:** 10.3390/foods14223861

**Published:** 2025-11-12

**Authors:** Ana Rosa Ramos-Polo, Ivan Luzardo-Ocampo, Sandra Navarro-Gallón, Silvia A. Quijano, Sandra Sulay Arango-Varela

**Affiliations:** 1Grupo de Investigación en Neurociencias y Envejecimiento (GISAM), Facultad de Ciencias de la Salud, Corporación Universitaria Remington, Calle 51 No. 51-27, Antioquia, Medellín 050012, Colombia; ana.ramos@uniremington.edu.co; 2Tecnologico de Monterrey, Institute for Obesity Research, Ave. Eugenio Garza Sada 2501 Sur, Col.: Tecnológico, Monterrey 64700, NL, Mexico; ivanluzardo@tec.mx; 3Tecnologico de Monterrey, School of Engineering and Sciences, Ave. Eugenio Garza Sada 2501 Sur, Col.: Tecnológico, Monterrey 64700, NL, Mexico; 4CECOLTEC Research Group, CECOLTEC Services S.A.S., Carrera 43A No. 18 Sur-135, Antioquia, Medellin 050012, Colombia; snavarro@cecoltecservices.com; 5Facultad de Ciencias Básicas, Universidad Santiago de Cali, Cali 760035, Colombia; 6Grupo de Investigación en Ecología y Conservación de la Biodiversidad (EcoBio), Cali 760001, Colombia; 7Grupo de Investigación e Innovación Biomédica, Facultad de Ciencias Exactas y Aplicadas, Instituto Tecnológico Metropolitano (ITM), Calle 73 No. 76A-354 Vía al Volador, Antioquia, Medellín 050012, Colombia; 8Grupo de Investigación Biología Médica, Facultad de Ciencias Exactas y Aplicadas, Instituto Tecnológico Metropolitano (ITM), Calle 73 No. 76A-354 Vía al Volador, Antioquia, Medellín 050012, Colombia

**Keywords:** Andean berry (*Vaccinium meridionale* Swartz), antioxidant capacity, bioactive compounds, functional foods, nutraceutical applications

## Abstract

Andean berry (*Vaccinium meridionale* Swartz) is a species of berry mostly exclusive to the Andean ecosystems, mainly present in Colombia, Venezuela, Peru, and Jamaica, where it grows between 2000 and 3000 m.a.s.l. Although most of the fruit is harvested naturally, limited fruit production significantly restricts large-scale farming and sales. Most research on phytochemicals from this berry has focused on polyphenolic compounds, particularly anthocyanins such as cyanidin-3-*O*-galactoside and delphinidin-3-*O*-hexoside. These compounds have significant antioxidant potential and require appropriate post-harvest handling to preserve their stability and biological functionality. A systematic literature search was conducted covering studies from January 2000 to January 2025 across Scopus, PubMed, Web of Science, and Google Scholar. Evidence from original research includes chemical analyses, in vitro biological activity, in vivo effects in animal models, and clinical studies. Although findings suggest antiproliferative, chemoprotective, and cardioprotective properties, current evidence remains largely preclinical, and clinical validation is urgently needed. Despite its promise, challenges persist in standardizing cultivation, scaling production, and optimizing post-harvest. The berry has been incorporated into food products, but further research is essential to support its transition from experimental use to validated clinical applications.

## 1. Introduction

*Vaccinium meridionale* Swartz, commonly known as Andean berry, “mortiño”, or “agraz”, is a tetraploid species belonging to the Ericaceae family [1]. Research has emphasized a notable polyphenolic content, including anthocyanins, and its resulting antioxidant capacity, making it a functional food with potential nutraceutical value. The Andean berry belongs to the *Vaccinium* genus, which comprises approximately 450 species distributed worldwide, known for their rich polyphenol content and health benefits [2]. *V. meridionale* is mainly found in the Andean region of South America, especially in Colombia, Venezuela, and Peru, where it thrives in high mountain cloud forests and paramo as a scrub at altitudes ranging from 1000 to 3000 m above sea level. In Colombia, its distribution is vast, with the highest number of records in the Andean region of the country (Antioquia, Boyacá, and Cundinamarca states), and it is often found together with other *Vaccinium* species, such as V. *floribundum* and *V. corymbosum* [3]. This plant is well-adapted to cold, humid mountain habitats. Its habitat ranges from isolated populations in high-montane cloud forests to subparamo environments, where it grows as a shrub reaching up to 3.5 m in height. Its altitude range is 1000-2800 meters above the sea level [4]. It is noteworthy that flowering material has been collected in almost all months of the year, and mature fruits have been collected at different times, such as January–February, July–August, and November–December [5].

Morphologically (Figure 1), *V. meridionale* shows high polymorphism in qualitative traits and considerable variation in quantitative traits, both of which are crucial for taxonomic and agronomic purposes [1]. A single-nucleotide polymorphism (SNP) analysis, first conducted in 2025, identified 12910 SNPs, revealing moderate polymorphisms in *V. meridionale* populations from Colombia, with excess heterozygosity and low genetic differentiation among the country’s central regions. This examination of genetic diversity also identified three distinct subpopulations, but further studies are needed to confirm whether these correspond to distinct varieties [6].

Among the proposals of descriptors to study the variability of *V. meridionale,* a maturation scale has been proposed, ranging from 0 (green fruits) to 5 (dark purple fruits) [7]. For the highest maturation stage, the fruits are spherical berries when ripe, ranging from 5 to 20 mm in diameter and weighing from 2.6 to 6.8 g. These fruits are characterized by a lignified endocarp and a thick outer skin, reflecting their high antioxidant content [4]. At advanced stages of ripening, the inner surfaces of the loculi begin to accumulate pigment, suggesting that selected crosses may enhance flesh pigmentation. The plant exhibits abundant and concentrated flowering, with clustered inflorescences containing 15 to 25 flowers, which is advantageous for mechanical harvesting. It has been observed that this berry has potential for hybridization with other *Vaccinium* species, such as cranberry (*V. vitis-idaea*) and American cranberry (*V. macrocarpon*), producing more fertile hybrids with intermediate morphology and good fertility. This hybrid potential expands the applicability of conventional breeding programs [8].

In Colombia, *V. meridionale* has two harvest seasons per year: the first in April and May, and the second from September to December, with the latter being more abundant [9]. The ideal growing conditions are at altitudes between 2000 and 3800 m, with rainfall between 958 and 1350 mm, temperatures ranging from 13.5 to 22.3 °C, solar radiation between 16.1 and 21.3 MJ/m^2^, and soils with a pH of 4.4 to 5.4, rich in organic matter, loose, and porous. The seeding soil consists of a mixture of soil, sand, and commercial mycorrhizae in a ratio of 7.5:1.5:1*. V. meridionale* is produced by a small shrub that can reach up to 4 m in height and 5 cm in diameter, with a rounded crown and new brown branches that later turn light green [2]. The fruit of the Andean berry is a rich source of nutrients. In 100 g of fruit are found: 84.2–87.9% water, 3.5–6.1% protein, 1.0–4.7% fat, 16.2–17.4% fiber, 14.4–15.8% dry matter, 1.8–2.2% ash, 2.9–3.1% ether extract, 42 Kcal, 30 IU vitamin A, 0.014 mg vitamin B1, 0024 mg vitamin B2, 012 mg vitamin B6, 12 mg vitamin C, 0.2 mg niacin, 12 mg pantothenic acid, 2 mg sodium, 72 mg potassium, 14 mg calcium, 6 mg magnesium, 0.5 mg manganese, 0.5 mg iron, 0.6 mg copper, 10 mg phosphorus, and 4 mg chlorine [10].

The interest in this berry and other species of the genus lies mainly in their potential health benefits due to their high content of bioactive phytonutrients, including phenolic compounds such as flavonoids (including anthocyanins), phenolic acids, lignans, and polymeric tannins, which contribute to their antioxidant, antihyperlipidemic, antiproliferative, and neuroprotective activities [11,12]. Andean berry is characterized by an elevated total phenolic content (609–758 mg gallic acid equivalents GAE/100 g). The total anthocyanin content varies from 228 to 329 mg C-3-G equivalents per 100 g of ripe fresh fruit [13,14,15,16]. Regarding its antioxidant capacity, it has been determined using evaluation methods such as the 2,2-azino-bis(3-ethyl-benzothiazoline-6-sulfonic acid) (ABTS) (45.5 µmol Trolox equivalents, TE/g fresh weight, FW), the ferric reducing antioxidant power (FRAP) (87.3 µmol TE/g FW), and oxygen radical absorption capacity (ORAC) (27,116 µmol TE/100 g FF) [2]. When comparing the content of total phenolic compounds of some *Vaccinium* berries, it has been reported that Agraz contains up to twice as many total phenolic compounds when compared to other berries, such as the Northern Highbush blueberry (*Vaccinium corymbosum*) (181–473 mg GAE /100 g), Andean blackberry (*Rubus glaucus* Benth) (171.45 mg GAE/100 g) [17], the Rabbiteye blueberry (*Vaccinium ashei*) (230–457 mg GAE/100 g), the Lowbush blueberry (*Vaccinium angustifolium*) (290–495 mg GAE/100 g) [18] or *V. myrtillus,* which is highly known for its antioxidant and nutraceutical properties in several models of disease at in vitro and in vivo levels [19]. These polyphenolic compounds, whose levels vary based on genetic and environmental factors, are metabolized and excreted in humans. However, a significant amount reaches the large intestine, where it interacts with the gut microbiota to produce compounds that enhance chemoprotective effects against cardiovascular disease and certain cancers [20,21].

Despite this berry’s beneficial properties, research on its efficient and sustainable production remains limited, underscoring the lack of preclinical and clinical studies validating its effects. A better understanding of its agronomic requirements is necessary to develop cultivation systems under suitable climatic conditions that maximize its genetic potential [5]. However, morphological and genetic variability within wild populations suggests broad potential for selection and genetic improvement, which could enhance cultivation and commercialization worldwide. In terms of conservation, it faces challenges due to insufficient harvesting practices and human pressure on its natural habitat, which could result in genetic erosion of the species.

Andean berry has gained significance in the market due to its nutraceutical properties and potential applications in food products, such as yogurts and jams, as well as in the cosmetic industry [22]. However, little research has been conducted on the inclusion of Andean berry as a functional ingredient in food applications, and most reports have explored its biological properties in vitro or in vivo. Among food products, powdered Andean berry fruit has been suggested for preparing ice cream to improve its antioxidant capacity and techno-functional properties [23]. Particularly for this food development, Andean berry-based ice creams increased in hardness after 30 days of storage due to a decrease in overrun and no color changes, potentially thanks to the color protection offered by antioxidant compounds in the berry.

Another way Andean berry has been explored is its pomace, which accounts for roughly 20% of the fruit’s weight, is a valuable by-product that can serve as a natural colorant in Greek-style yogurt, enhancing its nutritional value, antioxidant capacity, and sensory appeal [24], and also complying with current regulations demanding natural colorants in the food industry [25]. Considering that *V. meridionale* pomace has a high fiber content, the addition of this by-product enhanced the physicochemical properties of the yogurt by preventing syneresis through additional hydrogen-bond formation, thereby increasing water-holding capacity [26]. Moreover, the novel pigmentation characteristic imparted by berries to fermented dairy products improves sensory acceptance, resulting in higher ratings for Andean berry pomace-added yogurts compared to a commercial control without berry addition [24].

Despite the high potential for food products manufacturing and the inclusion of *V. meridionale* in the list of accepted species in the U.S. market since 2006, there is no extensive commercialization of the raw or transformed fruit [3,27]. The high price this berry can reach makes its consumption mainly among higher socioeconomic groups. Stores and companies that sell and process the fruit require high-quality standards, such as clean, whole berries free of pest or disease damage, with a minimum size of 6–8 mm, rounded, and in optimal ripeness for fresh eating or processing. Yet, meeting these requirements is challenging because it is a wild plant, and the product is collected using traditional methods, such as collecting in nearby forests where it grows, identifying productive plants, and local harvesting and selection, which limit wide-scale commercialization [2,9].

The Andean berry remains a versatile and promising species with significant potential in both agriculture and the food industry. Its morphological traits, along with its high levels of bioactive compounds and the elevated number of flowers per branch, which is useful for mechanical harvesting, make it a strong candidate for breeding programs [8]. Research into its nutraceutical properties, coupled with efforts to develop sustainable cultivation practices, is essential for maximizing its potential in global fruit and nutraceutical markets. To encourage ongoing research and the dissemination of updated information on the Andean berry, this systematic review aims to present and discuss its biological properties, whether in whole or as a food matrix, with a particular focus on polyphenolic compounds, including anthocyanins, since these are the most relevant and extensively studied phytochemicals in *V. meridionale.*

## 2. Materials and Methods

Following the Preferred Reporting Items for Systematic Reviews and Meta-Analyses (PRISMA) guidelines, a systematic literature review was conducted to identify relevant studies on *Vaccinium meridionale* published between January 2000 and January 2025. The search was conducted in the Scopus, PubMed, Web of Science, and Google Scholar databases using combinations of the Medical Subject Headings (MeSH) terms and keywords such as “*Vaccinium meridionale*,” “*Vaccinium meridionale* Swartz,” “bioactive compounds,” “phenolic compounds,” and “anthocyanins.” To improve the accuracy of the results, Boolean operators (AND/OR) were used during the search (Figure 2).

Only original studies addressing specific aspects of the species were included, such as chemical analyses of its composition using high-performance liquid chromatography (HPLC), in vitro evaluations of biological activity, or in vivo research conducted in animal or human models. Considering the limited geographical distribution of *Vaccinium meridionale* and the scarce international research available, it was decided to also incorporate literature published in Spanish, in addition to articles in English, to cover regional scientific production.

During the selection process, duplicates, abstracts without access to the full text, reviews, and studies that did not present experimental data were excluded. Article selection and data extraction were performed independently by two reviewers.

## 3. Polyphenolic Compounds’ Composition of *Vaccinium meridionale*

### 3.1. Overall Composition

In recent years, scientific interest in this fruit has increased significantly due to its unique profile of polyphenolic compounds and antioxidant capacity, making it a valuable addition to the field of functional foods [28]. This interest mainly arises from the fruit’s high content of bioactive compounds, including various types of polyphenolic compounds (phenolic acids, flavonoids—such as anthocyanins—and tannins), which are associated with numerous health benefits. These compounds have shown antioxidant and anti-inflammatory effects and may play a vital role in preventing chronic diseases linked to oxidative stress, such as cardiovascular disease, cancer, and diabetes [29,30].

The next section reviews relevant studies investigating the composition of polyphenolic compounds in *V. meridionale*, its antioxidant capacity, and the effects of processing on its bioactive components. The data are summarized in Table 1 and Table 2, providing a comprehensive overview of the levels of total phenolic compounds and antioxidant capacity measured by various methods. This allows for a clear assessment of the nutritional and functional potential of this food. Gaviria-Montoya et al. [31] were the first to report the total anthocyanin content in fresh fruit, indicating 200.6 ± 10.2 mg of cyanidin-3-*O*-glucoside per 100 g of fresh fruit and 609 ± 31 mg GAE/100 g of total phenolic compounds. These results suggest that the Andean berry is an excellent source of antioxidants, with levels comparable to those of other berries such as bilberry (*Vaccinium myrtillus*) (538.6 µg cyanidin-3-glucoside/g), blueberry (*Vaccinium corymbosum*) (51.76 mg cyanidin-3-glucoside/g), maqui berry (*Vaccinium floribundum*) (1445–2240 mg/L) [32], which are known for their high levels of these bioactive compounds [33].

Garzón et al. [13] expanded on this research and reported even higher levels of anthocyanins in fresh fruit, with 329.0 ± 28.0 mg of C3G per 100 g and a total phenolic content of 758.6 ± 62.3 GAE per 100 g. Additionally, these authors identified cyanidin-3-*O*-galactoside as the main anthocyanin and chlorogenic acid as the most abundant phenolic acid. This study also indicates that the diversity of bioactive compounds in Andean berries extends beyond anthocyanins to include other phenolic compounds, such as chlorogenic and caffeic acids, which have demonstrated antioxidant and anti-inflammatory effects in preclinical studies. Besides anthocyanins, other studies have reported flavonoids such as quercetin and rutin, which also possess antioxidant properties and may influence inflammatory responses. For example, Arango et al. [37] reported the presence of several phenolic acids in freeze-dried Andean berry juice, including gallic acid, chlorogenic acid, and *p*-coumaric acid, which are known to inhibit lipid peroxidation and prevent cell damage caused by reactive oxygen species (ROS). The high antioxidant capacity of Andean berry has been demonstrated in several studies using different analytical methods, such as oxygen radical absorbance capacity (ORAC), ferric reducing antioxidant power (FRAP), and the 2,2-diphenyl-1-picrylhydrazyl (DPPH) radical inhibition

Lopera et al. [38] used the differential pH method to measure total monomeric anthocyanins in juice and Andean berry concentrate. Found that the total anthocyanin levels in the juice were 1224 mg/L (2.5 °Brix), while in the concentrate they reached 953.7 mg/L (19.5 °Brix). These levels are significantly higher than those reported for fruit juices like strawberry (*Fragaria vesca* L.) and pomegranate (*Punica granatum*), which contain approximately 55.7 mg/L and 250.87 mg/L of total anthocyanins, respectively. These findings highlight the potential of Andean berries as a rich source of antioxidants in liquid form, making them excellent candidates for the development of functional beverages with nutraceutical properties.

In another study, Estupiñan-Amaya et al. [36] assessed the total polyphenol content in freeze-dried Andean berry juices, finding elevated levels of total phenolic compounds (2032.5 ± 41.7 mg GAE/L) and total monomeric anthocyanins (371.5 ± 20.1 mg C3G/L). These results support the hypothesis that Andean berries have a notably higher antioxidant capacity than other berries, suggesting that regular consumption could significantly reduce oxidative stress in the body. Garzón et al. [35] reported 5794.3 ± 169.1 mg GAE/100 g of fresh fruit in *V. meridionale* pomace, indicating that this by-product is a potential source of antioxidant compounds. These values surpass those of other well-known *Vaccinium* species, such as *Vaccinium myrtillus* (1116.24 mg GAE/100 g). Berry pomace is an interesting by-product that could be explored from these fruits, since it is often rich in dietary fiber (up to 50% of the total dry weight) [39], also includes polyphenolic compounds, and can be used in a wide range of functional ingredients to improve their antioxidant properties or provide novel sensory characteristics [40]. These findings suggest that Andean berry by-products, like pomace, could be a valuable and cost-effective source of antioxidant compounds for use in the food and pharmaceutical industries. A list of representative polyphenolic compounds found in *V. meridionale* is provided in Table 2.

**Table 2 foods-14-03861-t002:** Main identified phenolic compounds in *V. meridionale* and their chemical structures.

Phenolic Compound	Chemical Structure	References
Cyanidin-3-*O*-glucoside	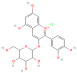	[13,31,35,41]
Cyanidin-3-*O*-galactoside	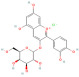	[13,35,38]
Cyanidin-3-arabinoside	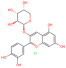	[13,41]
Chlorogenic acid	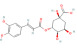	[13,35,42,43,44]
Caffeic acid	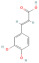	[13,35,43,45]
*p*-coumaric acid	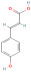	[42,44,46]
Ferulic acid	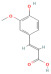	[44,45]
Quercetin	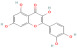	[13,35,45,46]
Rutin	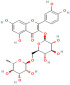	[44,45,46]

The compound structures were extracted from the National Center for Biotechnology Information (PubMed Compound Database).

### 3.2. Effect of Processing on the Content of Polyphenolic Compounds

Andean berry processing significantly affects the content of bioactive compounds, as shown in Table 3, especially total phenolic compounds and anthocyanins, which contribute to its antioxidant capacity. Fresh Andean berries serve as a reference, showing an extensive range of polyphenolic compounds, including anthocyanins, which may be due to the heterogeneity of reports reporting values per FW or dry weight [13,31,35,36,38]. These compounds are known for their strong antioxidant properties; however, no studies using colorimetric or fluorometric methods have been reported for the fresh fruit [35,36,38,47]. Nonetheless, the levels of these bioactive compounds tend to decrease over time or with specific processing techniques, highlighting the need to regulate these processes [48].

The freeze-drying process is one of the most effective methods for preserving phenolic compounds in fruit [49]. This technique is beneficial because it reduces the loss of bioactive compounds by removing water without using excessive heat, thereby maintaining the molecular structure of antioxidants. In contrast, freezing, although a useful method for long-term storage, has been shown to significantly reduce total phenolic compounds and anthocyanin content compared with fresh fruit [50]. Although freezing is gentler than thermal methods, such as pasteurization, ice crystal formation can damage fruit cells, promote oxidation, and reduce the levels of bioactive compounds. Furthermore, the antioxidant capacity of frozen Andean berries is not always evaluated with ORAC and FRAP assays, which limits comprehensive comparison with other preservation methods.

Andean berry juice is another important fruit-derived product that retains high levels of total phenolic compounds and anthocyanins, though to a lesser degree than fresh or freeze-dried fruit. The reported values, which are close to those of fresh fruit, suggest that the bioactive compounds have been successfully preserved. However, the processing involved in producing the juice may affect the stability of certain compounds, especially during heating, thereby reducing their antioxidant capacity. Nevertheless, the antioxidant activity, as measured by ORAC, is 24 µmol TE)/g, which remains significant from a nutraceutical perspective [33]. The juicing process may make antioxidants more readily available and easier to consume, though some polyphenols are inevitably lost.

It is notable that alternative techniques, such as aqueous extraction and maceration, have been observed to exert variable effects on the concentration of existing bioactive compounds. However, aqueous extraction has proven to be an effective technique for concentrating total phenolic compounds in this berry. This technique has been shown to enhance the concentration of total phenolic compounds, especially at higher temperatures. However, the duration of the extraction process and the thermal conditions can degrade some sensitive compounds, including certain flavonoids [51]. In contrast, maceration is less effective at concentrating total phenolic compounds and total monomeric anthocyanins [38], but it enables the extraction of more specific compounds, such as chlorogenic acid. While these methods are less effective than freeze-drying or juicing at preserving overall antioxidant levels, they may be useful in situations where targeted compounds are needed.

**Table 3 foods-14-03861-t003:** Effect of processing methods on total phenolic compounds, including total monomeric anthocyanins, and antioxidant capacity, in Andean berry (*Vaccinium meridionale* Swartz).

Sample Matrix	Processing Method	Total Phenolic Compounds	Total Monomeric Anthocyanins	Antioxidant Capacity	Impact on Selected Polyphenolic Compounds	References
Fruit	Unprocessed fruit	609 ± 31 mg GAE/100 g	200.60 ± 10.20 mg C3G/100 g	DPPH: 2404 ± 120 μM Trolox/100 g fresh fruit. ABTS: 8694 ± 435 μM Trolox/100 g fresh fruit. FRAP: 581 ± 29 mg ascorbic acid/100 g fresh fruit.	Chlorogenic acid, quercetin, and other flavonoids are present in significant concentrations.	[13,31]
Fruit	Lyophilization (freeze-drying)	1046.01–2546 mg GAE/100 g	82.64–150.70 mg C3G/100 g	ORAC: 41775.20 ± 6168.20 mmol Trolox/100 g lyophilized extract. FRAP: 5.10 mmol FeSO_4_/g.	Preserves most of the phenolic compounds but reduces the total monomeric anthocyanins content by 63.3%.	[49,50]
Fruit	Freezing	1046.01 ± 26.95 mg GAE/100 g	82.64 mg C3G/100 g	-	May slightly decrease phenolic compounds (including anthocyanins) compared to fresh fruit.	[31]
Extract	Aqueous extraction	4409.78 ± 63.05 mg GAE/100 g	106.57 mg C3G/100 g	-	Increased extraction of total phenolic compounds (including anthocyanins) improves the bioavailability of these compounds.	[44,52]
Extract	Maceration	86 ± 4 mg GAE/100 g	--	ABTS: 3.8 ± 0.30 µmol Trolox Eq/mL DPPH: 22.90 ± 5.40 µmol Trolox Eq/mL	Maceration can extract various phenolic compounds, and high levels of chlorogenic acid have been reported.	[49]
Fruit	Fruit juice	2032.50 ± 41.70 mg GAE/L	371.5 ± 20.1 mg C3G/L	ORAC: 24.00 µmol TE/g.	Improve the bioactivity of antioxidants; increase the concentration of vitamin C and other flavonoids.	[24]
Fruit	Concentration	953.70 mg GAE/100 g	1224 mg C3G/100 g	-	Concentrating may increase antioxidant activity but decrease the levels of heat-sensitive compounds.	[50]
Fruit	Osmotic dehydration	692.70–47.40 mg GAE/100 g	-	ORAC (hydrophilic fraction): 11,490.80 ± 631.60 μmol TE/100 g FRAP: 4084.50 ± 106.1 μmol TE/100 g DPPH: 5731.60 ± 108.80 μmol TE/100 g	Decrease in polyphenolic compounds’ content, including anthocyanin concentration in the drying process	[51]
Fruit	Drying at different temperatures (40, 50, and 60 °C)	3.87–47.40 mg GAE/100 g	1.21–2.36 mg C3G/100 g		At 40°c, 24% of anthocyanins are retained, with a greater loss as the temperature increases.	[30]
Vinegar	Aqueous/methanolic extract/HCL	4409.78 ± 63.05 mg GAE/100 mL	106.57 ± 1.43 mg C3G/100 mL	FRAP: 476.84 ± 18.81 µmol TE/100 mL. ORAC: 113,176.15 ± 4527.04 µmol TE/100 mL. DPPH: 128.03 ± 3.84 µmolTE/100 g Fresh Weight.	Prolonged process, moderate phenolic compounds content	[43]
Extract	Ethanolic extract	83.98–167.90 mg GAE/100 g	29.08–747.77 mg C3G/100 g	DPPH: 1143.68 ± 3.87 ug Trolox/g ABTS: 253.27 ± 12.57 g ascorbic acid/kg ORAC 472.25 ± 6.08 mg TE/g, FRAP 1.98 ± 0.02 mol ferric sulfate/kg	High levels of quercetin and cyanidin derivatives	[53]
Fruit	Freeze-dried juice	9.80 ± 0.04 mg GAE/100 g	-	-	Contains condensed tannins and flavonoids.	[30,38]
Wine	Wine	5794.30 ± 169.10 mg GAE/100 g	-	-	Contains a higher concentration of phenols when compared to *V. myrtillus*	[37]

ABTS: 2,2′-azino-bis(3-ethylbenzothiazoline-6-sulfonic acid); C3G: Cyanidin-3-*O*-glucoside; DPPH: 2,2-diphenyl-1-picrylhydrazil; FRAP: Ferric reducing antioxidant power; ORAC: Oxygen radical absorbance capacity; TE: Trolox equivalents.

## 4. Biological Effects Evaluated on *Vaccinium meridionale*

This section details the biological effects of *V. meridionale* through various experimental approaches. It includes findings from studies assessing its matrix extracts in both in vitro and in vivo models, as well as its incorporation into food matrices and the related effects observed under similar conditions. These subsections aim to offer a comprehensive understanding of the berry’s functional properties, emphasizing its antioxidant, anti-inflammatory, and other health-related activities.

### 4.1. Matrix Extracts

In vitro studies on *V. meridionale* extracts have provided valuable evidence about their therapeutic potential, especially regarding their cytotoxic, antioxidant, and antimicrobial activities across various cell models. One of the earliest studies was conducted by Maldonado-Celis et al. [49], who assessed the cytotoxic effects of freeze-dried extracts of Andean berry on SW480 and SW620 human colorectal cancer cell lines (half-maximal inhibitory concentration, or IC_50_): 59.12 and 56.10 µg/mL, respectively. The data indicate that the compounds present in this berry exert a direct cytotoxic effect on colon cancer cells. Given the promising reduction in metabolic activity in cancer cells, the results suggest that compounds in the Andean berry may be involved in key molecular pathways linked to cell growth. As such, the authors suggested cell cycle arrest, which has been further demonstrated by Andean berry aqueous extracts as a food matrix (juice), where G2/M arrest has been observed, followed by a decrease in Caspase 3/7 activity, suggesting antiproliferative effects beyond the classical apoptotic mechanism [44].

In another study, Sequeda-Castañeda et al. [47] examined the protective effect of a berry extract against induced oxidative stress in HT1080 cells, a human fibrosarcoma cell line. No toxicity was observed after testing the extract (10, 50, and 100 µg/mL) under normal cell conditions. When oxidative stress and cell damage were induced using a ROS-generating compound (rotenone), 50 and 100 µg/mL protected the cells, indicating significant antioxidant capacity in vitro derived from *V. meridionale* compounds. Additionally, González et al. [50] evaluated the cytotoxic effects of methanolic extracts of this berry on leukemia cell lines, focusing particularly on MOLT4 and OCI-AML3 cells. This study is relevant because leukemia cells are often resistant to standard treatments. A noticeable decrease in cell viability was seen in OCI-AML3 cells after exposure to increasing doses of *V. meridionale* extract (10, 50, and 100 μg/mL), with a maximum inhibition of 23% at the highest concentrations. However, no significant reduction in viability was observed in MOLT4 cells, suggesting that the cytotoxic effect is selective and depends on the cancer cell type, as observed in other matrices. Compared with the standard drug doxorubicin, the effects of the Andean berry extract were milder, suggesting that although it may have anti-leukemic properties, its potency is lower than that of traditional chemotherapies. The authors suggest that the mechanism of inhibiting cell proliferation could involve apoptosis, possibly through modulation of the nuclear factor κB (NF-κB) and activator protein-1 (AP-1) pathways, or by inhibiting enzymes such as cyclooxygenase. However, the authors highlighted that further research is needed to fully understand these mechanisms [47].

In a separate study, Agudelo et al. [45] investigated the impact of the fermented non-digestible fraction of Andean berry juice on SW480 cells’ proliferation. The Andean berry juice fraction was produced after simulated gastrointestinal digestion and colonic fermentation involving a human inoculum to mimic the gastrointestinal environment. Extracts at 15 and 18% *v*/*v* significantly reduced cell proliferation (*p* < 0.05), compared to untreated cells (IC_50_: 19.32% *v*/*v*). The chromatographic analysis of the fraction revealed the presence of gallic acid, chlorogenic acid, ellagic acid, rutin, and various complex sugars, which could jointly contribute to the observed biological activity. Furthermore, the aqueous extract of Andean Berry, rich in phenolic compounds, has been reported to exhibit antiproliferative activity in SW480 and SW620 colon cancer cells. The treatment arrested the cell cycle in S and G2/M phases (SW480) and in G0/G1 (SW620), inducing apoptosis. In SW620, DNA fragmentation was observed (SubG0/G1), whereas in SW480, the S-phase arrest was due to replicative damage. No mitochondrial depolarization or increase in ROS was observed, suggesting mitochondrial-independent apoptosis [54]. Also, the aqueous extract of Andean Berry was found to potentiate the antiproliferative effect of 5-fluorouracil (5-FU)+ leuprolide (LEU) and 5-FU + LEU + oxaliplatin (OXA) in SW480 and SW620 colorectal cancer cells. In SW620, it reduced migration and invasion, indicating an anti-metastatic effect, accompanied by decreased matrix metalloproteinase-9 (MMP-9). Although it did not affect drug-induced changes in cell adhesion, it did inhibit processes related to metastasis [55]. These results are consistent with previous evidence from other anthocyanin-rich berries, which increase the efficacy of chemotherapeutics, regulate proteins involved in adhesion and invasion, such as the intercellular adhesion molecule 1 (ICAM-1) and VCAM-1, and modulate metalloproteinase activity [56]. Additionally, Agudelo [57] investigated the impact of the same fermented fraction on the HT29 colon cancer cell line, observing a dose-dependent reduction in cell viability and an IC_50_ of 24.69%. Moreover, the treatment resulted in increased lactate dehydrogenase (LDH) release, a marker of cellular damage, indicating disruption of cell membrane integrity. TdT-mediated dUTP Nick-End labeling (TUNEL) assays, which detect apoptosis, confirmed that the fraction also promoted programmed cell death in HT29 cells. Finally, the antioxidant properties of the fraction were confirmed by a decrease in 8-isoprostane levels in cells after treatment with the fermented extract. Although no significant differences in catalase activity were observed, the enzyme superoxide dismutase (SOD) showed a 7.8% reduction, suggesting modulation of the endogenous antioxidant system. Garzón et al. [35] examined the antimicrobial activity of Andean berry pulp extract against pathogenic bacterial strains. Among the bacterial strains evaluated, *Staphylococcus aureus* exhibited the greatest sensitivity (IC_50_: 126 µg GAE/mL), compared to the *E. coli* O157 and *E. coli* ATCC 25922 strains (IC_50_: 334 and 528 µg GAE/mL, respectively). These findings align with previous studies on blueberries, which have shown that fractions rich in anthocyanins and other polyphenolic compounds can inhibit bacterial growth by disrupting cell membranes. The antimicrobial effect of the Andean berry thus reinforces its potential as a natural agent for preventing bacterial infections, particularly against antibiotic-resistant strains [58]. The authors primarily describe these effects as mediated by A-type proanthocyanidins, which not only impede bacterial adhesion to epithelial cells but also enhance the permeability of bacterial membranes, ultimately leading to their disintegration and cell death. The lower values observed for Gram-negative bacteria are due to the outer lipopolysaccharide (LPS) membrane surrounding the bacterial cell wall, which acts as an additional barrier to anthocyanins’ activity. This barrier primarily increases membrane permeability, leading to the leakage of electrolytes and even critical macromolecules, disturbing the internal cell balance [59].

In vivo studies on *V. meridionale* extracts indicate potential therapeutic benefits across a broad range of conditions, particularly cardiovascular and metabolic diseases. One of the most significant studies was conducted by Lopera et al. [38], who assessed the cardioprotective effects of a fermented non-alcoholic extract of Andean berry in rat hearts subjected to ischemia–reperfusion injury. This experimental model simulates damage during myocardial infarction, in which blood flow is interrupted and subsequently restored, leading to additional damage from ROS overproduction. The findings demonstrated a notable improvement in myocardial systolic and diastolic function in hearts treated with the extract. An increase in left ventricular developed pressure was observed, along with an improvement in the maximum rate of increase in ventricular pressure (+dP/dtmax) during reperfusion, indicating increased contractility of the treated heart. Furthermore, a reduction in left ventricular end-diastolic pressure was observed, indicating that the extract reduced myocardial stiffness during diastole, thereby enhancing the heart’s relaxation capacity. These results are particularly promising, as they suggest that fermented Andean berry extract could protect cardiac tissue from reperfusion-induced damage. This may be achieved by inducing endothelial nitric oxide synthase (eNOS) expression via the protein kinase (Akt) signaling pathway.

More recently, Shen et al. [60] also investigated the cardioprotective effects of the extract in an animal model of ischemia–reperfusion injury. The extract was fermented and then vacuum-evaporated to concentrate its bioactive compounds. A notable decrease in ROS production and lipid peroxidation (LPO), two pivotal markers of oxidative stress, was observed. Furthermore, treatment with the berry extract led to a notable increase in SOD and catalase activity, suggesting enhanced antioxidant defense mechanisms. Besides, there was a reduction in serum enzyme levels, including serum glutamic oxaloacetic transaminase (SGOT), serum glutamic pyruvic transaminase (SGPT), creatine kinase (CK), and LDH. Additionally, the study demonstrated elevated expression of Akt and eNOS proteins, suggesting that the extract may modulate pivotal pathways involved in cell protection and nitric oxide-mediated vasodilation. These findings reinforce the hypothesis that *V. meridionale* has significant cardioprotective potential. The authors recommended further in vivo studies in larger animal models or clinical trials to confirm these effects in a clinical setting.

In a 2020 study, Colorado et al. [61] investigated the impact of encapsulating grape extracts in non-ionic niosomes, a technology designed to improve the bioavailability of bioactive compounds, such as anthocyanins. In this study, a functional beverage containing encapsulated Andean berry anthocyanins was developed and administered to a mouse model of diet-induced obesity to assess its antihyperglycemic effects. The results indicated that the animals tolerated the beverage well, with no apparent toxicity observed during the experiment. Furthermore, mice that consumed the beverage showed a significant improvement in blood glucose regulation, demonstrating a strong antihyperglycemic effect. Since the supercritical fluid extraction primarily yielded C3G, cyanidin-3-arabinoside, cyanidin-3-galactoside, and delphinidin-3-hexoside, the authors attributed the observed biological effects to these anthocyanins. Therefore, these compounds have been reported to reverse insulin resistance, regulate GLUT4 expression and translocation, and increase peroxisome proliferator-activated receptor gamma (PPARγ). An effect also seen in pigmented food products containing anthocyanins and related polyphenolic compounds [62]. This finding is particularly important because obesity and type 2 diabetes are closely linked, and controlling blood glucose levels is essential for preventing complications associated with these conditions. The study emphasizes the potential of encapsulated Andean berry anthocyanins as a natural strategy to enhance blood glucose regulation and suggests that this technology could be used to develop functional products for individuals at risk of developing type 2 diabetes.

### 4.2. Food Matrices

In vitro studies examining the effects of *V. meridionale*-based food matrices have shown significant cytotoxic and antiproliferative activity against cancer cell lines, such as colon adenocarcinoma (SW480), as well as anti-inflammatory and antioxidant properties. In the study conducted by Zapata-Vahos et al. [42], the effect of an aqueous extract, which can be considered a “black tea” of Andean berry leaves, on SW480 cell proliferation was assessed. After 72 h of treatment, black tea caused a notable reduction in cell growth, with a 58.2% decrease at 20 μg/mL. A green tea prototype derived from Andean berry leaves showed a 49.6% reduction under similar conditions. Although both extracts displayed cytotoxicity, green tea demonstrated greater efficacy, with an IC_50_ of 26.3 μg/mL compared to 36.0 μg/mL for black tea. These results suggest that the bioactive compounds in Andean berry leaves may have antiproliferative potential, especially those associated with green tea, possibly due to a higher concentration of phenolic compounds or flavonoids compared to black tea.

In a study conducted by Franco et al. [50], the effect of nectars made with freeze-dried Andean berries on SW480 cells was examined. The results showed a dose-dependent increase in antiproliferative activity, indicating the potential of freeze-dried berries as a natural anti-cancer agent. The IC_50_ values after 48 and 72 h of exposure were 1.12 and 0.4 g/mL for one type of nectar and 0.60 g/mL for the other. This suggests that Andean berry nectars can exert a strong cytotoxic effect, decreasing cell viability in a time- and dose-dependent manner. The ability of Andean berry extracts to inhibit cell growth suggests they could be used as functional ingredients in products aimed at preventing or fighting colon cancer.

In a similar study, Agudelo et al. [63] investigated the effects of freeze-dried Andean berry juice (ABJ) on SW 480 cells. Treating cells with ABJ concentrations of 3% to 10% for 24, 48, and 72 h resulted in significant growth inhibition, which depended on both concentration and exposure time. The results showed that ABJ was particularly effective at 8% and 10%, with IC_50_ values of 19%, 8%, and 3% for 24, 48, and 72 h, respectively. Additionally, the study found that ABJ activated caspase-3, a key enzyme in the apoptotic process, and increased phosphorylation of p53 at Ser15, a marker of cellular damage and cell cycle regulation. It was also observed that ABJ caused a significant increase in ROS levels at concentrations above 6%, indicating that its cytotoxicity is mediated by oxidative stress and apoptosis induction. These findings support the idea that the compounds in V. meridionale can trigger programmed cell death mechanisms, highlighting its potential as a cancer treatment agent.

Conversely, Zapata et al. [43] studied the effects of vinegar made from this berry on the same SW480 cells. The vinegar significantly inhibited cell growth, reducing proliferation by 6.6–22.5% across concentrations from 20 μg/mL to 200 μg/mL. The estimated IC_50_ was 536 μg/mL, much higher than the 139.1 μg/mL reported for a fermented Andean berry beverage. This suggests that Andean berry vinegar has a weaker effect on inhibiting cancer cell growth than other fermented products. Still, it shows a noticeable antiproliferative effect, which may be due to the vinegar’s antioxidant compounds.

A more recent study by Arango-Varela et al. [37,44,64] investigated the joint effects of freeze-dried Andean berry juice (ABJ) and aspirin on nitric oxide (NO) production and cell proliferation in SW480 cells and RAW 264.7 macrophages. The viability of macrophages was not affected by any of the ABJ concentrations. However, a dose-dependent reduction in NO production was observed, with a more pronounced effect when ABJ was combined with aspirin. This suggests that the bioactive components of Andean berry and aspirin may act jointly to modulate inflammation. Moreover, ABJ, alone or in combination with aspirin, significantly reduced intracellular ROS production in LPS-stimulated macrophages, a key marker of inflammation [64]. Mechanistically, the authors conducted a proteomic analysis comparing the LPS-treated macrophages alone, with ABJ, and the ABJ + Aspirin combination, and found that the combination promoted mechanisms more involved with a decrease in the interleukin-1-receptor (IL-1R) activation, decreased granulocyte-macrophage colony-stimulating factor (GM-CSF), and reduced C-X-C motif chemokine ligand 10 (CXCL10), suggesting anti-inflammatory mechanisms of regulating the recruitment of immune cells, thereby producing lower amount of the classical pro-inflammatory cytokines [65]. Conversely, in SW480 cells, ABJ treatment resulted in G0/G1 cell cycle arrest, accompanied by a notable decline in the G2/M subpopulation, suggesting an impediment to cell cycle progression. The combination of ABJ and aspirin demonstrated a more pronounced reduction in cell viability, indicating a synergistic effect in inhibiting tumor growth. These findings are consistent with increased cell accumulation in the sub-G1 phase, suggesting induction of apoptosis [37]. In a subsequent study, it was demonstrated that both ABJ and the combination of ABJ with aspirin induced cell death mechanisms that extended beyond apoptosis. The modulation of pathways such as necroptosis or the activation of specific receptors by freeze-dried Andean berry juice appears to induce alternative mechanisms of cell death, thereby opening new possibilities for its application in oncology treatments. The studies indicate that Andean berry, in its various forms (juice, nectar, and vinegar), could be incorporated into functional food matrices to capitalize on its antioxidant, antiproliferative, and anti-inflammatory properties, thereby facilitating the development of therapeutic strategies against cancer and other diseases associated with oxidative stress.

In vivo studies on the consumption of Andean berry-derived food matrices have demonstrated promising effects on key health markers, including oxidative stress, inflammation, and insulin resistance, as well as on the prevention of chronic diseases. A more detailed analysis of various research findings is presented below. Fantinelli et al. [66] studied the cardioprotective effects of a fermented extract of *V. meridionale* using a model of ischemia–reperfusion, a condition in which blood flow is temporarily interrupted and then restored, leading to oxidative damage in cardiac tissue. The results showed that consuming this fermented extract significantly reduced myocardial contractile dysfunction and lessened reperfusion-induced oxidative damage. This effect was attributed to the bioactive compounds in Andean berry, which may neutralize ROS generated during reperfusion. Interestingly, neither polyphenol-rich cocoa extract nor whole berry extract showed similar benefits under the same conditions, underscoring the value of fermented Andean berry in this specific context.

Espinosa-Moncada et al. [52] conducted a clinical study on women with metabolic syndrome who consumed a reconstituted, freeze-dried preparation of Andean berry for four weeks. Although there were no significant differences in high-density lipoprotein (HDL) cholesterol levels or markers of oxidative stress and inflammation compared with the placebo group, the study provided valuable information on the safety and tolerability of berry consumption in this high-risk population. While substantial improvements in the measured markers were not observed, this research lays a foundation for future studies to explore higher doses or combinations with other treatments to assess their long-term effects on metabolic conditions.

Galvis-Pérez et al. [67] examined the effects of consuming nectar from the Andean berry on total phenol levels and serum antioxidant capacity in individuals with metabolic syndrome. Although no statistically significant differences were found in the primary variables, gender-specific responses emerged. Women who showed increased phenol levels experienced a significant reduction in insulin resistance, while men who increased their antioxidant capacity saw a marked decrease in oxidized low-density lipoprotein cholesterol (LDL). These findings underscore the potential for gender-specific metabolic and antioxidant effects of Andean berry, highlighting the importance of considering biological sex when designing future nutritional interventions to maximize effectiveness. Similarly, Quintero-Quiroz et al. [68] investigated how consuming nectar from Andean berries influenced people at high risk of cardiovascular disease. Participants consumed the nectar for 4 weeks, with a placebo serving as the control. Although increases in serum total phenol levels and antioxidant capacity, measured by FRAP, were not statistically significant, gender-based analysis revealed that women experienced a 19% increase in antioxidant capacity, assessed by the DPPH method, after consuming Andean berries compared with the placebo. This suggests that women may have heightened sensitivity to the antioxidant properties of Andean berry, which could be important in preventing cardiovascular disease in this group. The authors caution, however, that these outcomes may depend on the timing of sample collection and the methods used to measure antioxidant capacity.

In a study conducted by Gallego-Peláez et al. [51], the impact of osmo-dried Andean berry consumption on subjects with obesity and overweight over 21 days was examined. The findings indicated a notable decline in plasma concentrations of several pro-inflammatory cytokines, such as interleukin-6 (IL-6), interleukin-1β (IL-1β), and tumor necrosis factor-α (TNF-α), suggesting that Andean berry ingestion may help regulate inflammatory status in this demographic. However, the impact on body mass index was minimal and not statistically significant, indicating that while Andean berry may improve certain inflammatory markers, its effect on body weight may be limited in the short term. The study emphasizes the need for longer-term studies to better evaluate the effects of Andean berries on obesity and other metabolic syndrome risk factors.

Arango-Varela et al. [37] investigated the chemopreventive effect of Andean berry juice (ABJ) in a colorectal cancer model. The study demonstrated that ABJ consumption reduced the formation of aberrant crypt foci (ACF), an early marker of colorectal carcinogenesis. The bioactive compounds present in ABJ acted as chemopreventive agents by inhibiting key pro-inflammatory pathways, such as NF-κB and PI3K-Akt, which promote cell growth and inflammation. Moreover, ABJ was observed to reduce the activation of pro-inflammatory cytokines, including TNF-α and IL-1β, in LPS-challenged macrophages, thereby supporting the notion that *V. meridionale* may serve as a preventive measure against the advancement of colorectal cancer.

In vivo studies on food matrices from Andean berries indicate that they can be classified as functional foods with multiple health benefits. The most significant effects include enhanced antioxidant capacity and modulation of inflammatory markers, particularly in populations at risk of chronic diseases such as metabolic syndrome and colorectal cancer. Therefore, future interventions should take this factor into account to maximize benefits. Although the results are promising, further long-term research is needed to verify these findings and explore their potential impact on a wider range of health conditions. Additionally, these studies suggest there may be gender-based differences in the response to consuming these berries.

Recently, Mashhour et al. [69] reported the results of a systematic review on the impact of Andean berry supplementation on inflammation-related and oxidative stress markers in patients with metabolic syndrome, including 6 studies that fully met the authors’ quality criteria. The review concluded that most of the observed effects align with the anti-inflammatory properties of *V. meridionale* bioactive compounds, which primarily modulate the canonical inflammation pathway (NF-κB) and suppress pro-inflammatory genes, such as inducible nitric oxide synthase (iNOS) and cyclooxygenase-2 (COX-2). The exact mechanisms by which *V. meridionale* compounds enhance the activity of antioxidant enzymes remain unclear, underscoring the need for further clinical studies to support their biological properties at the preclinical level. Yet, the restricted ecosystem in which Andean berries grow decreases the availability of plant material and hinders large-scale comparative studies, unlike the extensive knowledge available for other *Vaccinium* species cultivated in North America and Europe. This situation limits the species’ international visibility and fragments available information, despite evidence of its phenotypic plasticity and its high content of secondary metabolites with functional and biomedical potential.

In vitro and in vivo studies provide insights into the use of Andean berry extracts for the management of cardiovascular and metabolic diseases. The cardioprotective effects observed demonstrate this berry’s ability to reduce tissue damage caused by oxidative stress during ischemia–reperfusion events and its potential to improve glucose control in models of obesity and diabetes. Adding these extracts to functional products could provide a natural, safe alternative to help prevent and reduce the impact of these chronic diseases. Additionally, encapsulating bioactive compounds is a promising strategy to enhance the effectiveness of Andean berry extracts, thereby increasing their bioavailability and maximizing therapeutic benefits. These technological advances in delivering bioactive compounds are essential for ensuring that the benefits seen in preclinical studies are effectively translated into human disease treatments. A summary of the described biological effects at the in vitro and in vivo levels is presented in Table 4.

## 5. Conclusions

*V. meridionale*, commonly known as the Andean berry, is gaining recognition as a functional food rich in polyphenolic compounds, including anthocyanins, which are recognized as antioxidants and anti-inflammatory agents with a wide range of biological benefits. The Andean berry contains even more total phenolic compounds than other well-known berries, primarily anthocyanins, chlorogenic acid, and quercetin derivatives. Most preclinical evidence suggests the anti-proliferative, chemopreventive, cardioprotective, antioxidant, hypoglycemic, and antihyperlipidemic properties of *V. meridionale*, but much of this research still needs clinical validation. Unfortunately, *V. meridionale* cultivation requires greater standardization to ensure consistent quality, yield enough polyphenolic compounds, and proper post-harvest handling. Production scaling is needed to meet market demands. Regarding processing, most studies have focused on freeze-dried and osmo-dehydrated fruits or juice, avoiding thermal methods that could harm the targeted polyphenolic compounds. As a result, food applications of the Andean berry have been diverse, successfully incorporated into yogurts and even cosmetic products. However, further research is needed to explore the potential of pomace from the fruit after juice production. Overall, its chemical composition, biological properties, and applications in commercial products make *V. meridionale* a promising ingredient or full functional matrix rich in polyphenolic compounds, conferring many desired health benefits that could be highly attractive to both local and international markets.

## Figures and Tables

**Figure 1 foods-14-03861-f001:**
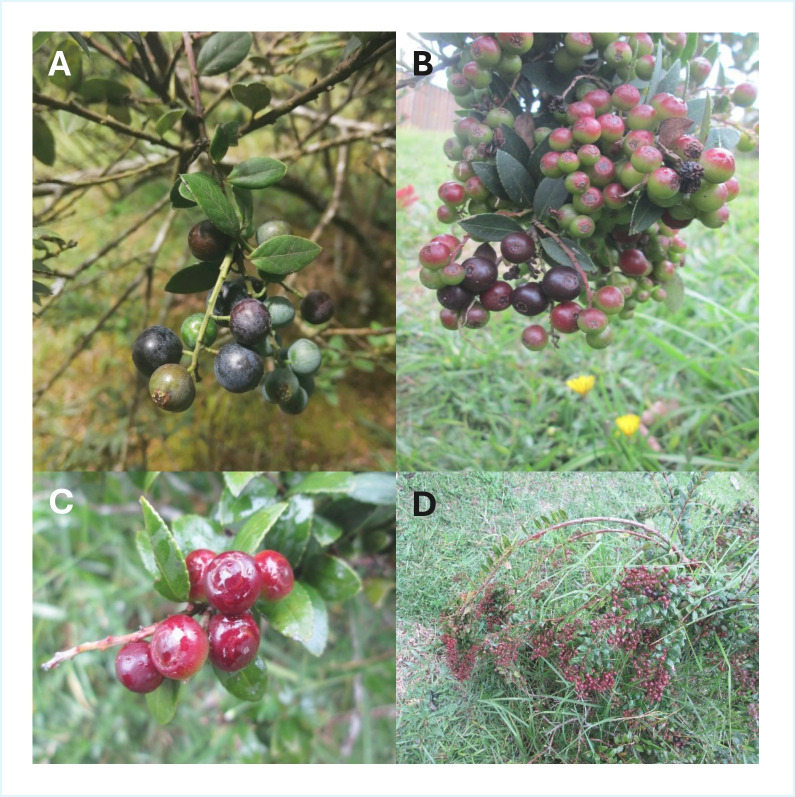
Morphological characteristics of *Vaccinium meridionale* Swartz. The image shows four photographs highlighting the distinctive features of *V.*
*meridionale*, commonly known as the Andean berry. (**A**) The plant’s overall morphology, including its branching structure, green leaves, and flowering parts; (**B**) fruits at different stages of ripeness, changing from green to deep purple as they mature; (**C**) close-up of the berries, emphasizing their small size, smooth texture, and glossy appearance; (**D**) the plant’s natural habitat.

**Figure 2 foods-14-03861-f002:**
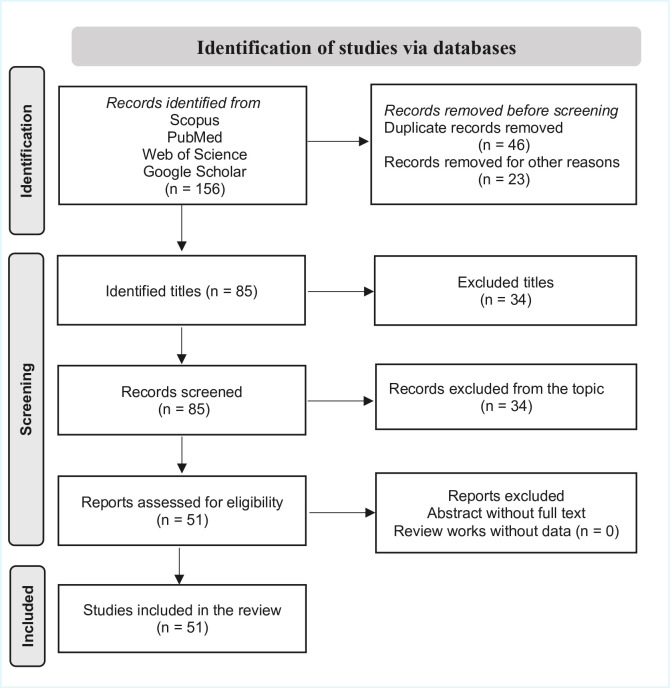
PRISMA flow diagram of the study selection process.

**Table 1 foods-14-03861-t001:** Total phenolic compounds and total monomeric anthocyanins content in *V*. *meridionale* Swartz.

Bioactive Compounds	Concentration Range	References
Total monomeric anthocyanins (mg C3G/100 g)	41.90–747.60	[13,31,34,35,36]
Total phenolic compounds (mg GAE/100 g)	141.20–1311.80	

C3G: Cyanidin-3-*O*-glucoside; GAE: gallic acid equivalents.

**Table 4 foods-14-03861-t004:** Biological effects from Andean berry (*Vaccinium meridionale* Swartz) fruit and derived matrices.

Evaluated Matrix	Biological Model Used	Main Effects	References
In vitro studies			
Vinegar from *Vaccinium meridionale*	SW480 colon cancer cells	Antioxidant capacity (↑ DPPH, ↑ FRAP, ↑ ORAC); Antiproliferative/cytotoxic effects (↓ cell viability, IC_50_ = 536 µg/mL).	[43]
Pomace (phenolic extract)	Bacteria: *Staphylococcus aureus*, *E. coli*	Antimicrobial activity (↑ inhibition zone vs. *S. aureus* > *E. coli*); Antioxidant capacity (↑ ORAC, ↑ ABTS).	[35]
Aqueous fruit extract (lyophilized)	SW480, SW620 colon cancer cell lines	Antioxidant (↑ total phenolic compounds, ↑ anthocyanins); cytotoxic/antiproliferative effects (↓ viability; IC_50_ ≈ 56–59 µg/mL).	[49]
Fermented non-digestible fraction of Andean berry juice	HT29 colon adenocarcinoma cells	Pro-apoptotic (↑ TUNEL positive cells, ↑ apoptosis markers); Antiproliferative (↓ viability, LC_50_ ≈ 24.7% v/v); Oxidative stress modulation (↓ SOD activity, ↑ 8-iso-PGF2α).	[57]
Fruit extracts (methanol) from several fruits, including *V. meridionale*	HT1080 fibroblasts	Antioxidant (↑ DPPH / ABTS / ORAC protection); Protective against rotenone-induced viability loss (↑ viability).	[47]
Alcoholic beverage extracts/dealcoholized fermented beverage	SW480 colon cancer cells	Antioxidant (↑ phenolics, anthocyanins); Antiproliferative (↓ viability; IC_50_ ≈ 139 µg/mL; Max inhibition 37.2% at 200 µg/mL).	[42]
Extracts (various: methanol, aqueous, juice)	OCI-AML3 and MOLT-4 leukemia cell lines	Antioxidant (↑ total phenolic compounds); Cytotoxic (↓ cell viability—modest vs. doxorubicin: ~23–24%)	[70]
Andean Berry Juice (ABJ)—digested/fermented fractions	SW480 colon cancer cells	Elevated bioaccessibility for gallic acid; Fermented fraction: strong antiproliferative effect (↓ viability, IC_50_ ≈ 19.3% *v*/*v*); markers.	[45]
*Vaccinium meridionale* extract adsorbed on bacterial nanocellulose	SW480, SW620 cancer cells and HaCaT keratinocytes	Cytotoxic (↓ viability in cancer lines; selectivity index shows less effect on HaCaT); Controlled release profile demonstrated (encapsulation protects in gastric pH).	[71]
Aqueous Andean berry extract	SW480, SW620 colon cancer cells	Antiproliferative (↓ viability; IC_50_ ≈ 19.2–23 mg/mL depending on cell line/time); Induced apoptosis (↑ Annexin V), cell-cycle arrest (S/G2-M or G0/G1 depending on the cell line); ROS modulation (↑ intracellular ROS, ↓ GSH).	[54]
Aqueous extract + 5-FU/leucovorin ± oxaliplatin	SW480 / SW620 colon cancer cells	Joint/co-adjuvant effect: ↓ viability and ↓ adhesion/migration/invasion in SW620; ↓ MMP-9 levels (↓ metastasis markers).	[55]
Andean Berry Juice + Aspirin	RAW 264.7 LPS-stimulated macrophages	Anti-inflammatory (↓ NO production, ↓ ROS); ↓ IL-1β, ↓ MCP-1, ↓ GCSF observed; predicted interactions with CCR1/CCR5/NF-κB (in silico).	[64]
	SW480 colon cancer cells	Antiproliferative (↓ viability), modulation of ROS/GSH balance; epigenetic modulation noted as a potential mechanism.	[44]
	SW480 colon cancer cells	In vitro ↑ antiproliferative and pro-apoptotic effects (↑ apoptosis markers, ↑ cell cycle arrest).	[37]
Andean Berry Juice	SW480 colon cancer cells	Antiproliferative / pro-apoptotic (↓ proliferation; IC_50_ ≈ 8% *v*/*v*); ↑ Caspase-3 activity; ↑ ROS; ↓ GSH (oxidative stress-mediated apoptosis).	[63]
Green and black tea from *V. meridionale* leaves (infusions)	SW480 colon cancer cells	Antioxidant (↑ TEAC/DPPH/ORAC); Antiproliferative (↓ viability; green tea IC_50_ ≈ 26.3 µg/mL, black tea ≈ 36 µg/mL).	[72]
In vivo studies			
Andean Berry juice + Aspirin	Mice treated with AOM	Chemoprotective effects: preserved colonic architecture; ↓ aberrant crypt foci formation (↓ early markers of colon carcinogenesis).	[37]
*V. meridionale* anthocyanin-loaded niosomes (oral)	Diet-induced obesity (DIO) mouse model	Metabolic improvements: ↓ fasting glucose (↓), ↓ insulin resistance (↑ insulin sensitivity), ↓ body weight, and visceral fat; improved lipid markers.	[61]
*Vaccinium meridionale* Sw. extracts	Male albino Wistar rats, ischemia–reperfusion model	Cardioprotective / antioxidant: ↑ SOD and CAT (↑ antioxidant enzymes), ↓ lipid peroxidation (↓ MDA), ↑ eNOS and Akt expression (↑ protective signaling).	[60]
Fruit	Isolated rat hearts (Langendorff)/ex vivo perfusion	Mixed cardioprotective outcomes: *V. meridionale* shows antioxidant activity and protective effects in some protocols (↑ antioxidant capacity; variable cardioprotection vs. other extracts).	[66]
Clinical studies			
Andean Berry Juice	Healthy volunteers with dietary CRC risk factors (250 mL/day for 14 days)	↑ Plasma antioxidant capacity (↑ Trolox eq. mM); ↓ isoprostanes (↓ 8-iso); ↓ IL-6 (↓ pro-inflammatory cytokine).	[73]
Freeze-dried fruit.	Adults with cardiovascular risk factors	Increases in antioxidant capacity in serum (↑ DPPH scavenging in women); variable systemic phenol changes — some sex differences.	[74]
Osmo-dehydrated fruit.	Overweight/obese adults (35 g/day for 21 days)	↓ Pro-inflammatory biomarkers (↓ IL-6, ↓ IL-1β, ↓ TNF-α); ↑ plasma antioxidant capacity; small decrease in BMI (non-robust).	[51]
Nectar	Women with MetS (human)	↑ Serum antioxidant capacity (↑); ↓ urinary 8-OH-dG (↓ DNA oxidative damage); modest changes in inflammatory/adipocytokine markers.	[52]
	Men and women with metabolic syndrome (human)	↑ Antioxidant capacity; ↓hs-CRP (↓ inflammation); sex-specific improvements (women: ↓ insulin resistance; men: ↓ oxLDL).	[67]
Reconstituted freeze-dried fruit in water (200 mL)	Men and women with MetS (human)	No significant change in HDL function or major inflammation markers overall; correlations observed (↑ PON1 correlated with ↑ cholesterol efflux).	[75]

Arrows up indicate an increase and arrows down indicate a decrease of the indicated marker. 8-iso-PGF2α: 8-iso-prostaglandin F2α; 8-OH-dG: 8-hydroxy-2′-deoxyguanosine; Akt: Protein kinase B; ABTS: 2,2′-azino-bis(3-ethylbenzothiazoline-6-sulfonic acid); BMI: Body mass index; C3G: Cyanidin-3-*O*-glucoside; CAT: Catalase; CCR1: C-C chemokine receptor 1; CCR5: C-C chemokine receptor 5; DPPH: 2,2-diphenyl-1-picrylhydrazil; eNOS: endothelial nitric oxide synthase; FRAP: Ferric reducing antioxidant power; GCSF: granulocyte colony-stimulating factor; GSH: Reduced glutathione; HDL: High-density lipoprotein; hs-CRP: high sensitivity-C reactive protein; IC_50_: Half-maximal inhibitory concentration; IL-1β: Interleukin 1β; IL-6: Interleukin-6; LC_50_: Half-maximal lethal concentration; MDA: malondialdehyde; LPS: Lipopolysaccharide; MMP-9: matrix metalloproteinase-9; NF-κB: Nuclear factor-κB; NO: Nitric oxide; ORAC: Oxygen radical absorbance capacity; ox-LDL: Oxidized low-density lipoprotein cholesterol; ROS: Reactive oxygen species; SOD: Superoxide dismutase; TE: Trolox equivalents; TEAC: Trolox equivalents antioxidant capacity; TNF-α: Tumor necrosis factor α.

## Data Availability

The original contributions presented in this study are included in the article. Further inquiries can be directed to the corresponding author.

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
