# Peer review of "Polyphenolic Compounds from Andean Berry (*Vaccinium meridionale* Swartz) and Derived Functional Benefits: A Systematic and Updated Review"

_foods, 2025, doi:10.3390/foods14223861_

Round 1

Reviewer 1 Report

Comments and Suggestions for Authors

The introduction, objectives, materials and methods are correct.

With regard to the results, it should be noted that the term ‘phytochemical composition’ is used when only phenolic compounds are discussed, which could lead to confusion and suggest that other bioactive phytochemical compounds are present. Therefore, the title could also lead to confusion. The topic is not relevant to the scientific field, as it discusses phytochemicals when it only delves into phenolic compounds. The focus of the work should be improved, and more effort should be put into creating tables and comparisons with recent studies.

The conclusions are very general and need further work.

The format and bibliographical references are correct.

Author Response

Medellin (Antioquia, Colombia), 1st November, 2025

Dr. Maoling Tu

Dr. Lianliang Liu

Dr. Shuzhen Cheng

Guest Editors, Foods

Special Issue: Nutritional Food Components: Their Natural Sources, Functions, and Various Applications

Dear editors, here you will find the answer to each of the reviewers’ comments

Reviewer 1

  1. Reviewer: With regard to the results, it should be noted that the term ‘phytochemical composition’ is used when only phenolic compounds are discussed, which could lead to confusion and suggest that other bioactive phytochemical compounds are present. Therefore, the title could also lead to confusion. The topic is not relevant to the scientific field, as it discusses phytochemicals when it only delves into phenolic compounds. The focus of the work should be improved, and more effort should be put into creating tables and comparisons with recent studies.

Only phenolic compounds are discussed. Anthocyanins are phenolic compounds belonging to the flavonoid family.

  • Author’s response: Dear reviewer, thank you for your comments. We agree with the reviewer that there are other phytochemicals besides polyphenolic compounds, so we have changed the title to “polyphenolic compounds”. Moreover, we have checked the entire document to avoid the confusion indicated by the reviewer. We also agree with the reviewer that anthocyanins are part of flavonoids, a class of polyphenolic compounds. We have corrected the sections in the manuscript that could suggest that anthocyanins are a separate class of compounds. However, anthocyanins have been presented separately from polyphenolic compounds when there is a need to make a distinction (for methodological procedures) between “total phenolic compounds” (quantified by the Folin-Ciocalteu method) and “total monomeric anthocyanins” (quantified using a differential pH method). Please refer to the revised manuscript.

Revised manuscript:

  • Page 1, lines 28-30: “(…) Most research on phytochemicals from this berry has focused on polyphenolic compounds, particularly anthocyanins such as cyanidin-3-O-galactoside and delphinidin-3-O-hexoside(…)”.

  • Page 2, lines 47-49: “(…) Research has emphasized a notable polyphenolic content, including anthocyanins, and its resulting antioxidant capacity (…)”

  • Page 5, Line 176-177: “To encourage ongoing research and the dissemination of updated information on the Andean berry, this systematic review aims to present and discuss its biological properties, whether in whole or as a food matrix, with a particular focus on polyphenolic compounds, including anthocyanins, since these are the most relevant and extensively studied phytochemicals in meridionale (…)”.

  • Page 5, lines 177-178: “(…) due to its unique profile of polyphenolic compounds and antioxidant capacity”.

  • Page 6, lines 205-207: “In recent years, scientific interest in this fruit has increased significantly due to its unique profile of polyphenolic compounds and antioxidant capacity, making it a valuable addition to the field of functional foods”.

  • Page 6, lines 213-215: “The next section reviews relevant studies investigating the composition of polyphenolic compounds in meridionale, its antioxidant capacity, and the effects of processing on its bioactive components.”

  • Page 9, line 319-321: “Fresh Andean berries serve as a reference, showing an extensive range of polyphenolic compounds, including anthocyanins, which may be due to the heterogeneity of reports reporting values per FW or dry weight”.

  • Page 13, lines 486-488: “These findings align with previous studies on blueberries, which have shown that fractions rich in anthocyanins and other polyphenolic compounds can inhibit bacterial growth by disrupting cell membranes.”

  • Page 14, 542-544: “An effect also seen in pigmented food products containing anthocyanins and related polyphenolic compounds [62].”.

  • Page 20, lines 733-734: meridionale, commonly known as the Andean berry, is gaining recognition as a functional food rich in polyphenolic compounds”.

Regarding the scientific relevance of this review, our research group considers it a much-needed review, as we have been researching the Andean berry for more than 15 years, and there is no updated information on this promising fruit. Although we believe the reviewer's comment is valuable, we are particularly specialized in polyphenolic compounds, which are the most relevant phytochemicals in this berry and the most studied and sought compositional elements when investigating its biological properties and applications. Therefore, we have focused the review on polyphenolic compounds as a more natural source.

  1. Reviewer: The conclusions are very general and need further work.

  • Authors’ response: We have rewritten the conclusions. Please refer to the revised manuscript:

  • Page 20, Lines 733-751: meridionale, commonly known as the Andean berry, is gaining recognition as a functional food rich in polyphenolic compounds, including anthocyanins, which are recognized as antioxidants and anti-inflammatory agents with a wide range of biological benefits. The Andean berry contains even more total phenolic compounds than other well-known berries, primarily anthocyanins, chlorogenic acid, and quercetin derivatives. Most preclinical evidence suggests the anti-proliferative, chemopreventive, cardioprotective, antioxidant, hypoglycemic, and antihyperlipidemic properties of V. meridionale, but much of this research still needs clinical validation. Unfortunately, V. meridionale cultivation requires greater standardization to ensure consistent quality, yield enough polyphenolic compounds, and proper post-harvest handling. Production scaling is needed to meet market demands. Regarding processing, most studies have focused on freeze-dried and osmo-dehydrated fruits or juice, avoiding thermal methods that could harm the targeted polyphenolic compounds. As a result, food applications of the Andean berry have been diverse, successfully incorporated into yogurts and even cosmetic products. However, further research is needed to explore the potential of pomace from the fruit after juice production. Overall, its chemical composition, biological properties, and applications in commercial products make V. meridionale a promising ingredient or full functional matrix rich in polyphenolic compounds, conferring many desired health benefits that could be highly attractive to both local and international markets. ”

  1. Reviewer: The format and bibliographical references are correct.

  • Author’s response: We appreciate the reviewer’s comments.

  1. Reviewer: 1 does not appear.

Author’s response: Dear reviewer, we are sorry, this was probably due to a conversion error. Fig. 1 has been added to the revised manuscript (Page 3).

  1. Reviewer: Materials and Methods. Research articles of the last 25 years?

  • Author’s response: Dear reviewer, thank you for your comments. We have amended the Materials and Methods section. We have also added a Figure (Figure 2). Please refer to the revised manuscript.

Revised manuscript:

  • Page 5, lines 182-199: “Following the Preferred Reporting Items for Systematic Reviews and Meta-Analyses (PRISMA) guidelines, a systematic literature review was conducted to identify relevant studies on Vaccinium meridionale published between January 2000 and January 2025. The search was conducted in the Scopus, PubMed, Web of Science, and Google Scholar databases using combinations of the Medical Subject Headings (MeSH) terms and keywords such as “Vaccinium meridionale,” “Vaccinium meridionale Swartz,” “bioactive compounds,” “phenolic compounds,” and “anthocyanins.” To improve the accuracy of the results, Boolean operators (AND/OR) were used during the search (Figure 2).

Only original studies addressing specific aspects of the species were included, such as chemical analyses of its composition using high-performance liquid chromatography (HPLC), in vitro evaluations of biological activity, or in vivo research conducted in animal or human models. Considering the limited geographical distribution of Vaccinium meridionale and the scarce international research available, it was decided to also incorporate literature published in Spanish, in addition to articles in English, to cover regional scientific production.

During the selection process, duplicates, abstracts without access to the full text, reviews, and studies that did not present experimental data were excluded. Article selection and data extraction were performed independently by two reviewers.”

  1. Reviewer: Table 3, what are the units of ABTS and DPPH? (reference 43)

  • Authors’ response: Units have been added. Please refer to Revised Table 3.

  1. Reviewer: Section 4. Biological effects evaluated on Vaccinium meridionale: create a table with the most relevant data and effects.

  • Author’s response: We have added Table 4. Please refer to the revised manuscript on Page 18, line 718.

We appreciate the significant time and effort the reviewer has dedicated to improving our manuscript. We hope the corrections we have made address the reviewer’s comments.

Prof. Dr. Silvia Andrea Quijano-Pérez

Facultad de Ciencias Básicas

Universidad Santiago de Cali

Corresponding Author 1

Prof. Dr. Sandra Sulay Arango-Varela

Grupo de Investigación Biología Médica

Facultad de Ciencias Exactas y Aplicadas

Instituto Tecnológico Metropolitano.

Corresponding Author 2

Reviewer 2 Report

Comments and Suggestions for Authors

This review lacks innovation, shows insufficient depth of discussion, and its conclusions are not concise. Moreover, author did not demonstrate adequate rigor. A major revision be suggested.

1.Where is Section 5?

2.Where is the Figure 1?

3.Pay attention to formatting: e.g., the last paragraph of Section 4.1 and the conclusion.

4.The review systematically summarizes the botanical characteristics, chemical composition, and potential health benefits of Vaccinium meridionale, but it lacks a new research perspective or theoretical framework.

5.What are the criteria and number of references collected?

6.The paper mainly cites South American and a few international studies, but it lacks a comprehensive comparison with other Vaccinium species (e.g., blueberry, cranberry), which limits the ability to highlight the uniqueness of this berry.

7.In the sections on antioxidant, anti-inflammatory, and anticancer effects, the discussion mostly remains at the level of “previous studies have shown,” without sufficient depth.

8.Although some human trials are mentioned, their number is limited and the results are not significant. The authors failed to emphasize these shortcomings adequately.

9.The article mainly focuses on laboratory results, with little discussion of challenges for the food industry (e.g., stability, cost, regulation, consumer acceptance) in applications such as functional beverages or nutraceuticals.

10.There are many tables and datasets, but they are not sufficiently intuitive.

11.The conclusion largely repeats earlier sections, lacking a synthesis of key findings and clear directions for future research.

12.The English is generally readable, but some sentences are overly long, logically disjointed, and repetitive (e.g., in the section on antioxidant mechanisms, information is overly piled up). Language refinement and professional editing are recommended.

13.The paper does not summarize the limitations of current research.

14.Some references are outdated; more recent systematic reviews and clinical studies should be incorporated to improve the timeliness of the literature.

Author Response

Medellin (Antioquia, Colombia), 1st November, 2025

Dr. Maoling Tu

Dr. Lianliang Liu

Dr. Shuzhen Cheng

Guest Editors, Foods

Special Issue: Nutritional Food Components: Their Natural Sources, Functions, and Various Applications

Dear editors, here you will find the answer to each of the reviewers’ comments

Reviewer 2

  1. Reviewer: This review lacks innovation, shows insufficient depth of discussion, and its conclusions are not concise. Moreover, author did not demonstrate adequate rigor. A major revision be suggested.

Where is Section 5?

  • Author’s response: Dear reviewer, thank you for your comments. We have improved the manuscript’s quality and deepened the discussion in several sections. Please refer to the revised manuscript.

Revised manuscript

  • Pages 3-4, lines 111-129: “Andean berry is characterized by an elevated total phenolic content (609–758 mg gallic acid equivalents GAE/100 g). The total anthocyanin content varies from 228 to 329 mg C-3-G equivalents per 100 g of ripe fresh fruit [13–16]. Regarding its antioxidant capacity, it has been determined using evaluation methods such as the 2,2-azino-bis(3-ethyl-benzothiazoline-6-sulfonic acid) (ABTS) (45.5 µmol Trolox equivalents, TE/g fresh weight, FW), the ferric reducing antioxidant power (FRAP) (87.3 µmol TE/g FW), and oxygen radical absorption capacity (ORAC) (27,116 µmol TE/100 g FF) [2]. When comparing the content of total phenolic compounds of some Vaccinium berries, it has been reported that Agraz contains up to twice as many total phenolic compounds when compared to other berries, such as the Northern Highbush blueberry (Vaccinium corymbosum) (181-473 mg GAE /100 g), Andean blackberry (Rubus glaucus Benth) (171.45 mg GAE/100 g) [17], the Rabbiteye blueberry (Vaccinium ashei) (230–457 mg GAE/100 g), the Lowbush blueberry (Vaccinium angustifolium) (290–495 mg GAE/100 g) [18] or myrtillus, which is highly known for its antioxidant and nutraceutical properties in several models of disease at in vitro and in vivo levels [19]. These polyphenolic compounds, whose levels vary based on genetic and environmental factors, are metabolized and excreted in humans. However, a significant amount reaches the large intestine, where it interacts with the gut microbiota to produce compounds that enhance chemoprotective effects against cardiovascular disease and certain cancers [20,21].”

  • Page 5, Lines 182-199: “Following the Preferred Reporting Items for Systematic Reviews and Meta-Analyses (PRISMA) guidelines, a systematic literature review was conducted to identify relevant studies on Vaccinium meridionale published between January 2000 and January 2025. The search was conducted in the Scopus, PubMed, Web of Science, and Google Scholar databases using combinations of the Medical Subject Headings (MeSH) terms and keywords such as “Vaccinium meridionale,” “Vaccinium meridionale Swartz,” “bioactive compounds,” “phenolic compounds,” and “anthocyanins.” To improve the accuracy of the results, Boolean operators (AND/OR) were used during the search (Figure 2).

Only original studies addressing specific aspects of the species were included, such as chemical analyses of its composition using high-performance liquid chromatography (HPLC), in vitro evaluations of biological activity, or in vivo research conducted in animal or human models. Considering the limited geographical distribution of Vaccinium meridionale and the scarce international research available, it was decided to also incorporate literature published in Spanish, in addition to articles in English, to cover regional scientific production.

During the selection process, duplicates, abstracts without access to the full text, reviews, and studies that did not present experimental data were excluded. Article selection and data extraction were performed independently by two reviewers.

  • Page 8, Lines 345-355: “It is notable that alternative techniques, such as aqueous extraction and maceration, have been observed to exert variable effects on the concentration of existing bioactive compounds. However, aqueous extraction has proven to be an effective technique for concentrating total phenolic compounds in this berry. This technique has been shown to enhance the concentration of total phenolic compounds, especially at higher temperatures. However, the duration of the extraction process and the thermal conditions can degrade some sensitive compounds, including certain flavonoids [51]. In contrast, maceration is less effective at concentrating total phenolic compounds and total monomeric anthocyanins [38], but it enables the extraction of more specific compounds, such as chlorogenic acid. While these methods are less effective than freeze-drying or juicing at preserving overall antioxidant levels, they may be useful in situations where targeted compounds are needed.”

  • Pages 13, Lines 457-472: “Furthermore, the aqueous extract of Andean Berry, rich in phenolic compounds, has been reported to exhibit antiproliferative activity in SW480 and SW620 colon cancer cells. The treatment arrested the cell cycle in S and G2/M phases (SW480) and in G0/G1 (SW620), inducing apoptosis. In SW620, DNA fragmentation was observed (SubG0/G1), whereas in SW480, the S-phase arrest was due to replicative damage. No mitochondrial depolarization or increase in ROS was observed, suggesting mitochondrial-independent apoptosis [54]. Also, the aqueous extract of Andean Berry was found to potentiate the antiproliferative effect of 5-fluorouracil (5-FU)+ leuprolide (LEU) and 5-FU+LEU+ oxaliplatin (OXA) in SW480 and SW620 colorectal cancer cells. In SW620, it reduced migration and invasion, indicating an anti-metastatic effect, accompanied by decreased matrix metalloproteinase-9 (MMP-9). Although it did not affect drug-induced changes in cell adhesion, it did inhibit processes related to metastasis [55]. These results are consistent with previous evidence from other anthocyanin-rich berries, which increase the efficacy of chemotherapeutics, regulate proteins involved in adhesion and invasion, such as the intercellular adhesion molecule 1 (ICAM-1) and VCAM-1, and modulate metalloproteinase activity [56].”

  • Page 13, lines 493-498: “The lower values observed for gram-negative bacteria are due to the outer lipopolysaccharide (LPS) membrane surrounding the bacterial cell wall, which acts as an additional barrier to anthocyanins’ activity. This barrier primarily increases membrane permeability, leading to the leakage of electrolytes and even critical macromolecules, disturbing the internal cell balance [59].”

  • Page 14, lines 538-542: “Since the supercritical fluid extraction primarily yielded C3G, cyanidin-3-arabinoside, cyanidin-3-galactoside, and delphinidin-3-hexoside, the authors attributed the observed biological effects to these anthocyanins. Therefore, these compounds have been reported to reverse insulin resistance, regulate GLUT4 expression and translocation, and increase peroxisome proliferator-activated receptor gamma (PPARγ).”

  • Pages 15-16, lines 598-607: “Moreover, ABJ, alone or in combination with aspirin, significantly reduced intracellular ROS production in LPS-stimulated macrophages, a key marker of inflammation [64]. Mechanistically, the authors conducted a proteomic analysis comparing the LPS-treated macrophages alone, with ABJ, and the ABJ+Aspirin combination, and found that the combination promoted mechanisms more involved with a decrease of the interleukin-1-receptor (IL-1R) activation, decreased granulocyte-macrophage colony-stimulating factor (GM-CSF), and reduced C-X-C motif chemokine ligand 10 (CXCL10), suggesting anti-inflammatory mechanisms of regulating the recruitment of immune cells, thereby producing lower amount of the classical pro-inflammatory cytokines [65].”

  • Page 17, lines 691-705: “Recently, Mashhour et al. [69] reported the results of a systematic review on the impact of Andean berry supplementation on inflammation-related and oxidative stress markers in patients with metabolic syndrome, including 6 studies that fully met the authors' quality criteria. The review concluded that most of the observed effects align with the anti-inflammatory properties of meridionale bioactive compounds, which primarily modulate the canonical inflammation pathway (NF-κB) and suppress pro-inflammatory genes, such as inducible nitric oxide synthase (iNOS) and cyclooxygenase-2 (COX-2). The exact mechanisms by which V. meridionale compounds enhance the activity of antioxidant enzymes remain unclear, underscoring the need for further clinical studies to support their biological properties at the preclinical level. Yet, the restricted ecosystem in which Andean berries grow decreases the availability of plant material and hinders large-scale comparative studies, unlike the extensive knowledge available for other Vaccinium species cultivated in North America and Europe. This situation limits the species' international visibility and fragments available information, despite evidence of its phenotypic plasticity and its high content of secondary metabolites with functional and biomedical potential.

  1. Reviewer: Where is the Figure 1?

  • Author’s response: We are very sorry, Fig. 1 could be missing due to a conversion error. We have added it. Please refer to the revised manuscript on Page 3.

  1. Reviewer: Pay attention to formatting: e.g., the last paragraph of Section 4.1 and the conclusion.

  • Author’s response: The last paragraph before the conclusions is now the Table 5 footnote.

  1. Reviewer: The review systematically summarizes the botanical characteristics, chemical composition, and potential health benefits of Vaccinium meridionale, but it lacks a new research perspective or theoretical framework.

  • Author’s response: Dear reviewer, we have completed the materials and methods section, indicating that this was a systematic review. Moreover, there are no updated reviews in the last 5 years. Our research group is particularly committed to investigating the potential health benefits of this promising berry across several disease models and to developing novel food applications that include meridionale as an ingredient. We believe there is still a need to highlight the fruit’s biological properties and to conduct additional clinical studies. We have placed these perspectives in the conclusions. Please refer to the revised manuscript.

Revised manuscript:

  • Page 20, lines 733-751: meridionale, commonly known as the Andean berry, is gaining recognition as a functional food rich in polyphenolic compounds, including anthocyanins, which are recognized as antioxidants and anti-inflammatory agents with a wide range of biological benefits. The Andean berry contains even more total phenolic compounds than other well-known berries, primarily anthocyanins, chlorogenic acid, and quercetin derivatives. Most preclinical evidence suggests the anti-proliferative, chemopreventive, cardioprotective, antioxidant, hypoglycemic, and antihyperlipidemic properties of V. meridionale, but much of this research still needs clinical validation. Unfortunately, V. meridionale cultivation requires greater standardization to ensure consistent quality, yield enough polyphenolic compounds, and proper post-harvest handling. Production scaling is needed to meet market demands. Regarding processing, most studies have focused on freeze-dried and osmo-dehydrated fruits or juice, avoiding thermal methods that could harm the targeted polyphenolic compounds. As a result, food applications of the Andean berry have been diverse, successfully incorporated into yogurts and even cosmetic products. However, further research is needed to explore the potential of pomace from the fruit after juice production. Overall, its chemical composition, biological properties, and applications in commercial products make V. meridionale a promising ingredient or full functional matrix rich in polyphenolic compounds, conferring many desired health benefits that could be highly attractive to both local and international markets.”

  1. Reviewer: What are the criteria and number of references collected?

  • Author’s response: Thank you for your comments. We have added the methodology used to collect the references, in accordance with PRISMA guidelines. Please refer to the revised manuscript.

Revised manuscript:

  • Page 5, Lines 182-199: “Following the Preferred Reporting Items for Systematic Reviews and Meta-Analyses (PRISMA) guidelines, a systematic literature review was conducted to identify relevant studies on Vaccinium meridionale published between January 2000 and January 2025. The search was conducted in the Scopus, PubMed, Web of Science, and Google Scholar databases using combinations of the Medical Subject Headings (MeSH) terms and keywords such as “Vaccinium meridionale,” “Vaccinium meridionale Swartz,” “bioactive compounds,” “phenolic compounds,” and “anthocyanins.” To improve the accuracy of the results, Boolean operators (AND/OR) were used during the search (Figure 2).

Only original studies addressing specific aspects of the species were included, such as chemical analyses of its composition using high-performance liquid chromatography (HPLC), in vitro evaluations of biological activity, or in vivo research conducted in animal or human models. Considering the limited geographical distribution of Vaccinium meridionale and the scarce international research available, it was decided to also incorporate literature published in Spanish, in addition to articles in English, to cover regional scientific production.

During the selection process, duplicates, abstracts without access to the full text, reviews, and studies that did not present experimental data were excluded. Article selection and data extraction were performed independently by two reviewers.”

  1. Reviewer: The paper mainly cites South American and a few international studies, but it lacks a comprehensive comparison with other Vaccinium species (e.g., blueberry, cranberry), which limits the ability to highlight the uniqueness of this berry.

  • Author’s response: Dear reviewer, thank you for your comments. We have not included other berries, but we have included them marginally, since our main focus was Vaccinium meridionale. However, after reviewing the reviewer’s comments, we have added comparisons with other berries. Please refer to the revised manuscript.

Revised manuscript:

  • Pages 3-4, lines 111-129: “Andean berry is characterized by an elevated total phenolic content (609–758 mg gallic acid equivalents GAE/100 g). The total anthocyanin content varies from 228 to 329 mg C-3-G equivalents per 100 g of ripe fresh fruit [13–16]. Regarding its antioxidant capacity, it has been determined using evaluation methods such as the 2,2-azino-bis(3-ethyl-benzothiazoline-6-sulfonic acid) (ABTS) (45.5 µmol Trolox equivalents, TE/g fresh weight, FW), the ferric reducing antioxidant power (FRAP) (87.3 µmol TE/g FW), and oxygen radical absorption capacity (ORAC) (27,116 µmol TE/100 g FF) [2]. When comparing the content of total phenolic compounds of some Vaccinium berries, it has been reported that Agraz contains up to twice as many total phenolic compounds when compared to other berries, such as the Northern Highbush blueberry (Vaccinium corymbosum) (181-473 mg GAE /100 g), Andean blackberry (Rubus glaucus Benth) (171.45 mg GAE/100 g) [17], the Rabbiteye blueberry (Vaccinium ashei) (230–457 mg GAE/100 g), the Lowbush blueberry (Vaccinium angustifolium) (290–495 mg GAE/100 g) [18] or myrtillus, which is highly known for its antioxidant and nutraceutical properties in several models of disease at in vitro and in vivo levels [19]. These polyphenolic compounds, whose levels vary based on genetic and environmental factors, are metabolized and excreted in humans. However, a significant amount reaches the large intestine, where it interacts with the gut microbiota to produce compounds that enhance chemoprotective effects against cardiovascular disease and certain cancers [20,21].”

  1. Reviewer: In the sections on antioxidant, anti-inflammatory, and anticancer effects, the discussion mostly remains at the level of “previous studies have shown,” without sufficient depth.

  • Author’s response: Dear reviewer, we have improved the discussion of these sections. Please refer to the revised manuscript.

Revised manuscript:

  • Page 12, Lines 425-428: “As such, the authors suggested cell cycle arrest, which has been further demonstrated by Andean berry aqueous extracts as a food matrix (juice), where G2/M arrest has been observed, followed by a decrease in Caspase 3/7 activity, suggesting antiproliferative effects beyond the classical apoptotic mechanism [44].”

  • Page 13, Lines 457-472: “Furthermore, the aqueous extract of Andean Berry, rich in phenolic compounds, has been reported to exhibit antiproliferative activity in SW480 and SW620 colon cancer cells. The treatment arrested the cell cycle in S and G2/M phases (SW480) and in G0/G1 (SW620), inducing apoptosis. In SW620, DNA fragmentation was observed (SubG0/G1), whereas in SW480, the S-phase arrest was due to replicative damage. No mitochondrial depolarization or increase in ROS was observed, suggesting mitochondrial-independent apoptosis [54]. Also, the aqueous extract of Andean Berry was found to potentiate the antiproliferative effect of 5-fluorouracil (5-FU)+ leuprolide (LEU) and 5-FU+LEU+ oxaliplatin (OXA) in SW480 and SW620 colorectal cancer cells. In SW620, it reduced migration and invasion, indicating an anti-metastatic effect, accompanied by decreased matrix metalloproteinase-9 (MMP-9). Although it did not affect drug-induced changes in cell adhesion, it did inhibit processes related to metastasis [55]. These results are consistent with previous evidence from other anthocyanin-rich berries, which increase the efficacy of chemotherapeutics, regulate proteins involved in adhesion and invasion, such as the intercellular adhesion molecule 1 (ICAM-1) and VCAM-1, and modulate metalloproteinase activity [56].”

  • Page 12, Lines 493-498: “The lower values observed for gram-negative bacteria are due to the outer lipopolysaccharide (LPS) membrane surrounding the bacterial cell wall, which acts as an additional barrier to anthocyanins’ activity. This barrier primarily increases membrane permeability, leading to the leakage of electrolytes and even critical macromolecules, disturbing the internal cell balance [59].”

  • Page 14, Lines 538-544: “Since the supercritical fluid extraction primarily yielded C3G, cyanidin-3-arabinoside, cyanidin-3-galactoside, and delphinidin-3-hexoside, the authors attributed the observed biological effects to these anthocyanins. Therefore, these compounds have been reported to reverse insulin resistance, regulate GLUT4 expression and translocation, and increase peroxisome proliferator-activated receptor gamma (PPARγ). An effect also seen in pigmented food products containing anthocyanins and related polyphenolic compounds [62].”

  • Pages 15-16, Lines 598-607: “Moreover, ABJ, alone or in combination with aspirin, significantly reduced intracellular ROS production in LPS-stimulated macrophages, a key marker of inflammation [64]. Mechanistically, the authors conducted a proteomic analysis comparing the LPS-treated macrophages alone, with ABJ, and the ABJ+Aspirin combination, and found that the combination promoted mechanisms more involved with a decrease of the interleukin-1-receptor (IL-1R) activation, decreased granulocyte-macrophage colony-stimulating factor (GM-CSF), and reduced C-X-C motif chemokine ligand 10 (CXCL10), suggesting anti-inflammatory mechanisms of regulating the recruitment of immune cells, thereby producing lower amount of the classical pro-inflammatory cytokines [65].”

  1. Reviewer: Although some human trials are mentioned, their number is limited and the results are not significant. The authors failed to emphasize these shortcomings adequately.

  • Author’s response: Thank you for your comments, and we agree with the reviewer. Unfortunately, there are few clinical studies targeting Vaccinium meridionale, and we present the published ones in this manuscript. We have included a systematic review published this year to complement the results we have already presented. Please refer to the revised manuscript.

Revised manuscript:

  • Page 17, Lines 682-705: In vivo studies on food matrices from Andean berries indicate that they can be classified as functional foods with multiple health benefits. The most significant effects include enhanced antioxidant capacity and modulation of inflammatory markers, particularly in populations at risk of chronic diseases such as metabolic syndrome and colorectal cancer. Therefore, future interventions should take this factor into account to maximize benefits. Although the results are promising, further long-term research is needed to verify these findings and explore their potential impact on a wider range of health conditions. Additionally, these studies suggest there may be gender-based differences in the response to consuming these berries.

Recently, Mashhour et al. [69] reported the results of a systematic review on the impact of Andean berry supplementation on inflammation-related and oxidative stress markers in patients with metabolic syndrome, including 6 studies that fully met the authors' quality criteria. The review concluded that most of the observed effects align with the anti-inflammatory properties of V. meridionale bioactive compounds, which primarily modulate the canonical inflammation pathway (NF-κB) and suppress pro-inflammatory genes, such as inducible nitric oxide synthase (iNOS) and cyclooxygenase-2 (COX-2). The exact mechanisms by which V. meridionale compounds enhance the activity of antioxidant enzymes remain unclear, underscoring the need for further clinical studies to support their biological properties at the preclinical level. Yet, the restricted ecosystem in which Andean berries grow decreases the availability of plant material and hinders large-scale comparative studies, unlike the extensive knowledge available for other Vaccinium species cultivated in North America and Europe. This situation limits the species' international visibility and fragments available information, despite evidence of its phenotypic plasticity and its high content of secondary metabolites with functional and biomedical potential.”

  1. Reviewer: The article mainly focuses on laboratory results, with little discussion of challenges for the food industry (e.g., stability, cost, regulation, consumer acceptance) in applications such as functional beverages or nutraceuticals.

  • Author’s response: We agree with the reviewer, but we believe the article's main focus is not the challenges faced by the food industry, as the reviewer noted. Please refer to the revised manuscript.

Revised manuscript:

  • Page 4, Lines 139-169: “Andean berry has gained significance in the market due to its nutraceutical properties and potential applications in food products, such as yogurts and jams, as well as in the cosmetic industry [22]. However, little research has been conducted on the inclusion of Andean berry as a functional ingredient in food applications, and most reports have explored its biological properties in vitro or in vivo. Among food products, powdered Andean berry fruit has been suggested for preparing ice cream to improve its antioxidant capacity and techno-functional properties [23]. Particularly for this food development, Andean berry-based ice creams increased in hardness after 30 days of storage due to a decrease in overrun and no color changes, potentially thanks to the color protection offered by antioxidant compounds in the berry.

Another way Andean berry has been explored is its pomace, which accounts for roughly 20% of the fruit's weight, is a valuable by-product that can serve as a natural colorant in Greek-style yogurt, enhancing its nutritional value, antioxidant capacity, and sensory appeal [24], and also complying with current regulations demanding natural colorants in the food industry [25]. Considering that V. meridionale pomace has a high fiber content, the addition of this by-product enhanced the physicochemical properties of the yogurt by preventing syneresis through additional hydrogen-bond formation, thereby increasing water-holding capacity [26]. Moreover, the novel pigmentation characteristic imparted by berries to fermented dairy products improves sensory acceptance, resulting in higher ratings for Andean berry pomace-added yogurts compared to a commercial control without berry addition [24].

Despite the high potential for food products manufacturing and the inclusion of V. meridionale in the list of accepted species in the U.S. market since 2006, there is no extensive commercialization of the raw or transformed fruit [3,27]. The high price this berry can reach makes its consumption mainly among higher socioeconomic groups. Stores and companies that sell and process the fruit require high-quality standards, such as clean, whole berries free of pest or disease damage, with a minimum size of 6-8 mm, rounded, and in optimal ripeness for fresh eating or processing. Yet, meeting these requirements is challenging because it is a wild plant, and the product is collected using traditional methods, such as collecting in nearby forests where it grows, identifying productive plants, and local harvesting and selection, which limit wide-scale commercialization [2,9].

The Andean berry remains a versatile and promising species with significant potential in both agriculture and the food industry. Its morphological traits, along with its high levels of bioactive compounds and the elevated number of flowers per branch, which is useful for mechanical harvesting, make it a strong candidate for breeding programs [8]. Research into its nutraceutical properties, coupled with efforts to develop sustainable cultivation practices, is essential for maximizing its potential in global fruit and nutraceutical markets. To encourage ongoing research and the dissemination of updated information on the Andean berry, this systematic review aims to present and discuss its biological properties, whether in whole or as a food matrix, with a particular focus on polyphenolic compounds, including anthocyanins, since these are the most relevant and extensively studied phytochemicals in V. meridionale.

  1. Reviewer: There are many tables and datasets, but they are not sufficiently intuitive.

  • Author’s response: Dear reviewer, thank you for your comments. We have added an additional table (Table 4) that we believe summarizes the section on biological effects. We have corrected some sections of Table 3 to make it more understandable, particularly that anthocyanins are also polyphenolic compounds. We hope these amendments fulfill the reviewer’s request.

  1. Reviewer: The conclusion largely repeats earlier sections, lacking a synthesis of key findings and clear directions for future research.

  • Author’s response: We appreciate the reviewer’s comments. We have rewritten the conclusions. Please refer to the revised manuscript.

Revised manuscript:

  • Page 20, lines 733-751: meridionale, commonly known as the Andean berry, is gaining recognition as a functional food rich in polyphenolic compounds, including anthocyanins, which are recognized as antioxidants and anti-inflammatory agents with a wide range of biological benefits. The Andean berry contains even more total phenolic compounds than other well-known berries, primarily anthocyanins, chlorogenic acid, and quercetin derivatives. Most preclinical evidence suggests the anti-proliferative, chemopreventive, cardioprotective, antioxidant, hypoglycemic, and antihyperlipidemic properties of V. meridionale, but much of this research still needs clinical validation. Unfortunately, V. meridionale cultivation requires greater standardization to ensure consistent quality, yield enough polyphenolic compounds, and proper post-harvest handling. Production scaling is needed to meet market demands. Regarding processing, most studies have focused on freeze-dried and osmo-dehydrated fruits or juice, avoiding thermal methods that could harm the targeted polyphenolic compounds. As a result, food applications of the Andean berry have been diverse, successfully incorporated into yogurts and even cosmetic products. However, further research is needed to explore the potential of pomace from the fruit after juice production. Overall, its chemical composition, biological properties, and applications in commercial products make V. meridionale a promising ingredient or full functional matrix rich in polyphenolic compounds, conferring many desired health benefits that could be highly attractive to both local and international markets.”

  1. Reviewer: The English is generally readable, but some sentences are overly long, logically disjointed, and repetitive (e.g., in the section on antioxidant mechanisms, information is overly piled up). Language refinement and professional editing are recommended.

  • Author’s response: Thank you for your comments. We have improved the readability of several sections of the manuscript. Please refer to the revised document.

  1. Reviewer: The paper does not summarize the limitations of current research.

  • Author’s response: We have added the limitations. Please refer to the revised manuscript.

Revised manuscript:

  • Page 17, lines 697-705: “The exact mechanisms by which meridionale compounds enhance the activity of antioxidant enzymes remain unclear, underscoring the need for further clinical studies to support their biological properties at the preclinical level. Yet, the restricted ecosystem in which Andean berries grow decreases the availability of plant material and hinders large-scale comparative studies, unlike the extensive knowledge available for other Vaccinium species cultivated in North America and Europe. This situation limits the species' international visibility and fragments available information, despite evidence of its phenotypic plasticity and its high content of secondary metabolites with functional and biomedical potential.”

  1. Reviewer: Some references are outdated; more recent systematic reviews and clinical studies should be incorporated to improve the timeliness of the literature.

  • Authors’ response: Dear reviewer, we have not identified any additional clinical trials beyond those presented in this manuscript. We have added only 1 systematic review focused on the Andean berry, but no others were found.

We appreciate the significant time and effort the reviewer has dedicated to improving our manuscript. We hope the corrections we have made address the reviewer’s comments.

Prof. Dr. Silvia Andrea Quijano-Pérez

Facultad de Ciencias Básicas

Universidad Santiago de Cali

Corresponding Author 1

Prof. Dr. Sandra Sulay Arango-Varela

Grupo de Investigación Biología Médica

Facultad de Ciencias Exactas y Aplicadas

Instituto Tecnológico Metropolitano.

Corresponding Author 2

Reviewer 3 Report

Comments and Suggestions for Authors

This paper reviewed the composition and its functions of andean berry. The information provided can improve the knowledge of this plant food. There are flaws that need improve in this paper, the details are listed below.

  1. Figure 1 is not showed in the paper, the morphological pictures are needed.
  2. Is the andean berry have many varieties or types ? if have, does they have any differences ?
  3. Does the components of andean berry have differences when it harvest at different time ?
  4. The table 1 showed the content of phenolic and anthocyanins were laid in range so big that let the reader doubt the materials were same species, so please check again.
  5.  The part 4, a list in table about the biological effects could make this part more clear.

Author Response

Medellin (Antioquia, Colombia), 1st November, 2025

Dr. Maoling Tu

Dr. Lianliang Liu

Dr. Shuzhen Cheng

Guest Editors, Foods

Special Issue: Nutritional Food Components: Their Natural Sources, Functions, and Various Applications

Dear editors, here you will find the answer to each of the reviewers’ comments

Reviewer 3

  1. Reviewer: This paper reviewed the composition and its functions of andean berry. The information provided can improve the knowledge of this plant food. There are flaws that need improve in this paper, the details are listed below.

Figure 1 is not showed in the paper, the morphological pictures are needed.

  • Author’s response: Thank you for your comments on improving this manuscript. We have included Fig. 1; we apologize it was not included previously. Perhaps it was a mistake during the file conversion. Please refer to the revised manuscript.

  1. Reviewer: Is the andean berry have many varieties or types ? if have, does they have any differences ?

  • Author’s response: Dear reviewer, thank you for bringing this interesting question, which we have answered in the revised manuscript. In May 2025, the first genetic diversity study of Vaccinium meridionale was published, suggesting moderate genetic diversity in Colombian populations of meridionale and identifying 3 distinct subpopulations that have not yet been classified as “varieties”. Please refer to the revised manuscript.

Revised manuscript:

  • Page 2, lines 64-71: “Morphologically (Figure 1), meridionale shows high polymorphism in qualitative traits and considerable variation in quantitative traits, both of which are crucial for taxonomic and agronomic purposes [1]. A single-nucleotide polymorphism (SNP) analysis, first conducted in 2025, identified 12910 SNPs, revealing moderate polymorphisms in V. meridionale populations from Colombia, with excess heterozygosity and low genetic differentiation among the country's central regions. This examination of genetic diversity also identified three distinct subpopulations, but further studies are needed to confirm whether these correspond to distinct varieties [6].”

  1. Reviewer: Does the components of andean berry have differences when it harvest at different time ?

  • Author’s response: Certainly, these components can vary just like in other plants. A previous report identified several descriptors for this plant, including the colors of the mature fruit, which range from purplish to dark purple, violet, and black. The classification of the Andean berry's maturation stages has also been documented and assigned a scale from 0 to 5, along with their physicochemical parameters. However, there are no reports on changes in individual or specific compounds at each stage, though most studies focus on dark purple fruits, which clearly correspond to the final maturation stage. We have included this information in the revised manuscript and thank the reviewer for raising this important question, which has enriched the text. Although most research on Andean berries does not report the maturation stage, we believe that most authors use fruits at their highest stage since the 5th stage has very distinct colors.

Revised manuscript:

  • Page 2, lines 72-73: “Among the proposals of descriptors to study the variability of meridionale, a maturation scale has been proposed, ranging from 0 (green fruits) to 5 (dark purple fruits) [7]. ”

  1. Reviewer: The table 1 showed the content of phenolic and anthocyanins were laid in range so big that let the reader doubt the materials were same species, so please check again.

  • Author’s response: Dear reviewer, we have double-checked and corrected the values. Differences were originally due to some researchers reporting values in fresh weight or dry weight, but all measurements were conducted on the same fruit species. Unfortunately, the range of values is still large, but we are considering the values reported in the table. Please refer to the revised manuscript on Page 7, lines 226-230.

  1. Reviewer: The part 4, a list in table about the biological effects could make this part more clear.

  • Author’s response: Thanks, we have added Table 4. Please refer to the revised manuscript on Pages 18-19, Lines 718-720.

We appreciate the significant time and effort the reviewer has dedicated to improving our manuscript. We hope the corrections we have made address the reviewer’s comments.

Prof. Dr. Silvia Andrea Quijano-Pérez

Facultad de Ciencias Básicas

Universidad Santiago de Cali

Corresponding Author 1

Prof. Dr. Sandra Sulay Arango-Varela

Grupo de Investigación Biología Médica

Facultad de Ciencias Exactas y Aplicadas

Instituto Tecnológico Metropolitano.

Corresponding Author 2

Reviewer 4 Report

Comments and Suggestions for Authors

Dear Authors,

Thank you for your manuscript, “Phytochemical composition and potential health benefits of Andean berry (Vaccinium meridionale Swartz) as a functional food: a narrative and updated review.” We appreciate the considerable effort you have invested in compiling a comprehensive body of literature on this promising and under-researched Andean fruit. The topic is timely and holds significant interest for researchers and industry professionals in the fields of functional foods, nutraceuticals, and ethnobotany. Your work provides a valuable foundational resource by gathering extensive data on the berry’s phytochemical profile and preclinical biological activities.

However, while the manuscript has strong potential, it currently requires major revision to meet the standards for publication. The following general and specific comments are intended to guide you in significantly strengthening the paper. Please address each point thoroughly in your revision.

General Comments

  1. Despite being called a "narrative review," the manuscript mostly summarizes findings study by study, with little critical synthesis or interpretation. A narrative review should highlight trends, controversies, and gaps—not just list results. The paper needs a stronger analytical voice.
  2. The manuscript jumps from Section 4 to Section 6, leaving Section 5 entirely absent. This major omission suggests poor proofreading and makes the review incomplete. Section 5 should address challenges, future perspectives, or a synthesis of biological effects.
  3. The abstract, introduction, and conclusion overstate the health benefits of the berry, often presenting preclinical findings as if clinically validated. The lack of human trials and methodological limitations must be clearly acknowledged.
  4. The "Materials and Methods" is superficial—databases and keywords are listed, but there is no information on screening, article counts, inclusion/exclusion criteria, or data extraction. This lack of transparency undermines credibility.
  5. Content is repeated across sections, tables are inconsistently formatted, and some data lack units or context. This reduces clarity and readability.

Specific Comments

  • Abstract exaggerates clinical relevance; revise to emphasize that evidence is mostly preclinical, with clinical validation urgently needed.
  • Background on botany and distribution is useful but overly optimistic. Statements like "wild harvesting ensures quality" are unsubstantiated. Temper enthusiasm, stress the preliminary nature of findings, and frame the review around potential and gaps.
  • Expand substantially. Provide detailed search strategy (databases, strings), article counts at each stage (identified, screened, included, excluded with reasons), and data extraction methods. Transparency is essential.
  • Phytochemical Composition Tables are useful but poorly presented. Reformat Table 3 with consistent units, standardized decimals, and a column for "Sample Matrix" (fruit, juice, pomace, extract). Avoid repeating table data in the text—add interpretation (e.g., why pomace has higher phenolics, why values vary widely).
  • Biological Effects are too descriptive—just lists findings. Organize by activity (anticancer, cardioprotective, anti-diabetic, etc.), and within each, present in vitro → in vivo → human evidence. Critically evaluate concentrations, models, and study quality. Add a summary table (model, intervention, findings, limitations). Explicitly discuss the gap between promising preclinical data and weak human evidence.
  • Current version is overly promotional. Rewrite to reflect the evidence—highlight promising phytochemistry and preclinical results, but stress the absence of strong clinical support. End with a roadmap for research rather than marketing language.

Author Response

Medellin (Antioquia, Colombia), 1st november, 2025

Dr. Maoling Tu

Dr. Lianliang Liu

Dr. Shuzhen Cheng

Guest Editors, Foods

Special Issue: Nutritional Food Components: Their Natural Sources, Functions, and Various Applications

Dear editors, here you will find the answer to each of the reviewers’ comments

Reviewer 4

  1. Reviewer: Thank you for your manuscript, “Phytochemical composition and potential health benefits of Andean berry (Vaccinium meridionale Swartz) as a functional food: a narrative and updated review.” We appreciate the considerable effort you have invested in compiling a comprehensive body of literature on this promising and under-researched Andean fruit. The topic is timely and holds significant interest for researchers and industry professionals in the fields of functional foods, nutraceuticals, and ethnobotany. Your work provides a valuable foundational resource by gathering extensive data on the berry’s phytochemical profile and preclinical biological activities.

However, while the manuscript has strong potential, it currently requires major revision to meet publication standards. The following general and specific comments are intended to guide you in significantly strengthening the paper. Please address each point thoroughly in your revision.

Despite being called a "narrative review," the manuscript mostly summarizes findings study by study, with little critical synthesis or interpretation. A narrative review should highlight trends, controversies, and gaps—not just list results. The paper needs a stronger analytical voice.

  • Author’s response: We appreciate the reviewer’s comments and their help in improving the manuscript. We mistakenly interpreted the “narrative” attribute of this review as a systematic review, and the revised Section 2 of the revised manuscript now explains this (Page 5, lines 181-199). Moreover, we have added Fig. 1 on Page 3.

We agree with the reviewer that each section should take on a more critical tone. Hence, we have thoroughly improved the manuscript’s sections to make them more critical as suggested by the reviewer. Please refer to the revised version of the manuscript.

  1. Reviewer: The manuscript jumps from Section 4 to Section 6, leaving Section 5 entirely absent. This major omission suggests poor proofreading and makes the review incomplete. Section 5 should address challenges, future perspectives, or a synthesis of biological effects.

  • Author’s response: We have simplified the manuscript’s divisions, and there are currently 4 sections. We decided to avoid the latest section since the manuscript has already noted limitations, current research, and commercial needs.

  1. Reviewer: The abstract, introduction, and conclusion overstate the health benefits of the berry, often presenting preclinical findings as if clinically validated. The lack of human trials and methodological limitations must be clearly acknowledged.

  • Author’s response: We have moderated the tone of these sections. Please refer to the revised manuscript.

Revised manuscript

  • Page 2, lines 47-49: “Research has emphasized a notable polyphenolic content, including anthocyanins, and its resulting antioxidant capacity, making it a functional food with potential nutraceutical value.”

  • Page 2, lines 80-84: “It has been observed that this berry has potential for hybridization with other Vaccinium species, such as cranberry ( vitis-idaea) and American cranberry (V. macrocarpon), producing more fertile hybrids with intermediate morphology and good fertility. This hybrid potential expands the applicability of conventional breeding programs [8].”

  • Page 4, lines 125-129: “These polyphenolic compounds, whose levels vary based on genetic and environmental factors, are metabolized and excreted in humans. However, a significant amount reaches the large intestine, where it interacts with the gut microbiota to produce compounds that enhance chemoprotective effects against cardiovascular disease and certain cancers [20,21].”

  • Page 4, lines 141-143: “However, little research has been conducted on the inclusion of Andean berry as a functional ingredient in food applications, and most reports have explored its biological properties in vitro or in vivo.”

  • Page 4, lines 160-162: “Despite the high potential for food products manufacturing and the inclusion of meridionale in the list of accepted species in the U.S. market since 2006, there is no extensive commercialization of the raw or transformed fruit [3,27].”

  • Page 5, lines 170-176: “The Andean berry remains a versatile and promising species with significant potential in both agriculture and the food industry. Its morphological traits, along with its high levels of bioactive compounds and the elevated number of flowers per branch, which is useful for mechanical harvesting, make it a strong candidate for breeding programs [8]. Research into its nutraceutical properties, coupled with efforts to develop sustainable cultivation practices, is essential for maximizing its potential in global fruit and nutraceutical markets.”

  • Page 20, lines 733-751: meridionale, commonly known as the Andean berry, is gaining recognition as a functional food rich in polyphenolic compounds, including anthocyanins, which are recognized as antioxidants and anti-inflammatory agents with a wide range of biological benefits. The Andean berry contains even more total phenolic compounds than other well-known berries, primarily anthocyanins, chlorogenic acid, and quercetin derivatives. Most preclinical evidence suggests the anti-proliferative, chemopreventive, cardioprotective, antioxidant, hypoglycemic, and antihyperlipidemic properties of V. meridionale, but much of this research still needs clinical validation. Unfortunately, V. meridionale cultivation requires greater standardization to ensure consistent quality, yield enough polyphenolic compounds, and proper post-harvest handling. Production scaling is needed to meet market demands. Regarding processing, most studies have focused on freeze-dried and osmo-dehydrated fruits or juice, avoiding thermal methods that could harm the targeted polyphenolic compounds. As a result, food applications of the Andean berry have been diverse, successfully incorporated into yogurts and even cosmetic products. However, further research is needed to explore the potential of pomace from the fruit after juice production. Overall, its chemical composition, biological properties, and applications in commercial products make V. meridionale a promising ingredient or full functional matrix rich in polyphenolic compounds, conferring many desired health benefits that could be highly attractive to both local and international markets.”

  1. Reviewer: The "Materials and Methods" is superficial—databases and keywords are listed, but there is no information on screening, article counts, inclusion/exclusion criteria, or data extraction. This lack of transparency undermines credibility.

  • Author’s response: Dear reviewer, we have improved this section. Please refer to the revised manuscript.

Revised manuscript:

  • Page 5, lines 182-200: “Following the Preferred Reporting Items for Systematic Reviews and Meta-Analyses (PRISMA) guidelines, a systematic literature review was conducted to identify relevant studies on Vaccinium meridionale published between January 2000 and January 2025. The search was conducted in the Scopus, PubMed, Web of Science, and Google Scholar databases using combinations of the Medical Subject Headings (MeSH) terms and keywords such as “Vaccinium meridionale,” “Vaccinium meridionale Swartz,” “bioactive compounds,” “phenolic compounds,” and “anthocyanins.” To improve the accuracy of the results, Boolean operators (AND/OR) were used during the search (Figure 2).

Only original studies addressing specific aspects of the species were included, such as chemical analyses of its composition using high-performance liquid chromatography (HPLC), in vitro evaluations of biological activity, or in vivo research conducted in animal or human models. Considering the limited geographical distribution of Vaccinium meridionale and the scarce international research available, it was decided to also incorporate literature published in Spanish, in addition to articles in English, to cover regional scientific production.

During the selection process, duplicates, abstracts without access to the full text, reviews, and studies that did not present experimental data were excluded. Article selection and data extraction were performed independently by two reviewers.”

  1. Reviewer: Content is repeated across sections, tables are inconsistently formatted, and some data lack units or context. This reduces clarity and readability.

  • Authors’ response: We have improved Table 3 and added Table 4; we believe the tables are consistently formatted. We are aware that the tables exceed the horizontal width, but all other MDPI manuscripts present tables in this format. We have tried to adjust the content to avoid repetition. Please refer to the revised manuscript.

  1. Reviewer: Abstract exaggerates clinical relevance; revise to emphasize that evidence is mostly preclinical, with clinical validation urgently needed.

  • Author’s response: Thanks, we have rewritten the abstract as suggested. Please refer to the revised manuscript.

Revised manuscript:

  • Page 1, lines 25-41: “Andean berry (Vaccinium meridionale Swartz) is a species of berry exclusive to high Andean ecosystems, mainly present in Colombia, Venezuela, Peru, and Jamaica, where it grows between 2000 and 3000 m.a.s.l. Although most of the fruit is harvested naturally, limited fruit production significantly restricts large-scale farming and sales. Most research on phytochemicals from this berry has focused on polyphenolic compounds, particularly anthocyanins such as cyanidin-3-O-galactoside and delphinidin-3-O-hexoside. These compounds have significant antioxidant potential and require appropriate post-harvest handling to preserve their stability and biological functionality. A systematic literature search was conducted covering studies from January 2000 to January 2025 across Scopus, PubMed, Web of Science, and Google Scholar. Evidence from original research includes chemical analyses, in vitro biological activity, and in vivo effects in animal models. Although findings suggest antiproliferative, chemoprotective, and cardioprotective properties, current evidence remains largely preclinical, and clinical validation is urgently needed. Despite its promise, challenges persist in standardizing cultivation, scaling production, and optimizing post-harvest. The berry has been incorporated into food products, but further research is essential to support its transition from experimental use to validated clinical applications.”

  1. Reviewer: Background on botany and distribution is useful but overly optimistic. Statements like "wild harvesting ensures quality" are unsubstantiated. Temper enthusiasm, stress the preliminary nature of findings, and frame the review around potential and gaps.

  • Author’s response: Thanks, we have improved the manuscript as suggested. Please refer to the revised manuscript.

Revised manuscript:

  • Page 1, lines 27-28: “Although most of the fruit is harvested naturally, limited fruit production significantly restricts large-scale farming and sales.”

  • Page 2, lines 47-49: “Research has emphasized a notable polyphenolic content, including anthocyanins, and its resulting antioxidant capacity, making it a functional food with potential nutraceutical value.”

  • Page 4, lines 132-135: “A better understanding of its agronomic requirements is necessary to develop cultivation systems under suitable climatic conditions that maximize its genetic potential [5]. However, morphological and genetic variability within wild populations suggests broad potential for selection and genetic improvement, which could enhance cultivation and commercialization worldwide”.

  • Page 4, lines 160-169: “Despite the high potential for food products manufacturing and the inclusion of meridionale in the list of accepted species in the U.S. market since 2006, there is no extensive commercialization of the raw or transformed fruit [3,27]. The high price this berry can reach makes its consumption mainly among higher socioeconomic groups. Stores and companies that sell and process the fruit require high-quality standards, such as clean, whole berries free of pest or disease damage, with a minimum size of 6-8 mm, rounded, and in optimal ripeness for fresh eating or processing. Yet, meeting these requirements is challenging because it is a wild plant, and the product is collected using traditional methods, such as collecting in nearby forests where it grows, identifying productive plants, and local harvesting and selection, which limit wide-scale commercialization [2,9].

  1. Reviewer: Expand substantially. Provide detailed search strategy (databases, strings), article counts at each stage (identified, screened, included, excluded with reasons), and data extraction methods. Transparency is essential.

  • Author’s response: We have expanded the Materials and Methods and added Fig. 1 for more transparency. Please refer to the revised manuscript.

  1. Reviewer: Phytochemical Composition Tables are useful but poorly presented. Reformat Table 3 with consistent units, standardized decimals, and a column for "Sample Matrix" (fruit, juice, pomace, extract). Avoid repeating table data in the text—add interpretation (e.g., why pomace has higher phenolics, why values vary widely).

  • Author’s response: Units have been added, and decimals standardized as suggested. We have avoided repeating values in the text. Please refer to the revised manuscript.

Revised manuscript:

  • Page 9, lines 319-321: “Fresh Andean berries serve as a reference, showing an extensive range of polyphenolic compounds, including anthocyanins, which may be due to the heterogeneity of reports reporting values per fresh weight or dry weight [13,31,35,36,38].”

  • Page 9, lines 338-341: “The reported values, which are close to those of fresh fruit, suggest successful preservation of the bioactive compounds. However, the processing involved in producing the juice may affect the stability of certain compounds, especially during heating, thereby reducing their antioxidant capacity.”

  • Page 9, lines 348-355: “This technique has been shown to enhance the concentration of total phenolic compounds, especially at higher temperatures. However, the duration of the extraction process and the thermal conditions can degrade some sensitive compounds, including certain flavonoids [51]. In contrast, maceration is less effective at concentrating total phenolic compounds and total monomeric anthocyanins [38], but it enables the extraction of more specific compounds, such as chlorogenic acid. While these methods are less effective than freeze-drying or juicing at preserving overall antioxidant levels, they may be useful in situations where targeted compounds are needed.”

  1. Reviewer: Biological Effects are too descriptive—just lists findings. Organize by activity (anticancer, cardioprotective, anti-diabetic, etc.), and within each, present in vitro → in vivo → human evidence. Critically evaluate concentrations, models, and study quality. Add a summary table (model, intervention, findings, limitations). Explicitly discuss the gap between promising preclinical data and weak human evidence.

  • Author’s response: Thank you for your comments. We decided not to make divisions by activity because reports for each activity are very scarce, leaving some activities with just 1 report. However, we added a Summary Table and expanded information on the presented evidence. Please refer to the revised manuscript.

Revised manuscript:

  • Page 12, lines 421-428: “The data indicate that the compounds present in this berry exert a direct cytotoxic effect on colon cancer cells. Given the promising reduction in metabolic activity in cancer cells, the results suggest that compounds in the Andean berry may be involved in key molecular pathways linked to cell growth. As such, the authors suggested cell cycle arrest, which has been further demonstrated by Andean berry aqueous extracts as a food matrix (juice), where G2/M arrest has been observed, followed by a decrease in Caspase 3/7 activity, suggesting antiproliferative effects beyond the classical apoptotic mechanism [44].”

  • Page 12, lines 440-442: “However, no significant reduction in viability was observed in MOLT4 cells, suggesting that the cytotoxic effect is selective and depends on the cancer cell type, as observed in other matrices.”

  • Page 13, lines 457-472: “Furthermore, the aqueous extract of Andean Berry, rich in phenolic compounds, has been reported to exhibit antiproliferative activity in SW480 and SW620 colon cancer cells. The treatment arrested the cell cycle in S and G2/M phases (SW480) and in G0/G1 (SW620), inducing apoptosis. In SW620, DNA fragmentation was observed (SubG0/G1), whereas in SW480, the S-phase arrest was due to replicative damage. No mitochondrial depolarization or increase in ROS was observed, suggesting mitochondrial-independent apoptosis [54]. Also, the aqueous extract of Andean Berry was found to potentiate the antiproliferative effect of 5-FU+LEU and 5-FU+LEU+OXA in SW480 and SW620 colorectal cancer cells. In SW620, it reduced migration and invasion, indicating an anti-metastatic effect, accompanied by decreased MMP-9. Although it did not affect drug-induced changes in cell adhesion, it did inhibit processes related to metastasis [55]. These results are consistent with previous evidence from other anthocyanin-rich berries, which increase the efficacy of chemotherapeutics, regulate proteins involved in adhesion and invasion, such as ICAM-1 and VCAM-1, and modulate metalloproteinase activity [56].”

  • Page 13, lines 488-498: “The antimicrobial effect of the Andean berry thus reinforces its potential as a natural agent for preventing bacterial infections, particularly against antibiotic-resistant strains [58]. The authors primarily describe these effects as mediated by A-type proanthocyanidins, which not only impede bacterial adhesion to epithelial cells but also enhance the permeability of bacterial membranes, ultimately leading to their disintegration and cell death. The lower values observed for gram-negative bacteria are due to the outer lipopolysaccharide (LPS) membrane surrounding the bacterial cell wall, which acts as an additional barrier to anthocyanins’ activity. This barrier primarily increases membrane permeability, leading to the leakage of electrolytes and even critical macromolecules, disturbing the internal cell balance [59].

  • Page 15, lines 518-521: “A notable decrease in ROS production and lipid peroxidation (LPO), two pivotal markers of oxidative stress, was observed. Furthermore, treatment with the berry extract led to a notable increase in SOD and catalase activity, suggesting enhanced antioxidant defense mechanisms.”

  • Page 14, lines 526-529: “These findings reinforce the hypothesis that meridionale has significant cardioprotective potential. The authors recommended further in vivo studies in larger animal models or clinical trials to confirm these effects in a clinical setting.”

  • Page 14, lines 538-549: “Since the supercritical fluid extraction primarily yielded cyanidin-3-glucoside, cyanidin-3-arabinoside, cyanidin-3-galactoside, and delphinidin-3-hexoside, the authors attributed the observed biological effects to these anthocyanins. Therefore, these compounds have been reported to reverse insulin resistance, regulate GLUT4 expression and translocation, and increase peroxisome proliferator-activated receptor gamma (PPARγ). An effect also seen in pigmented food products containing anthocyanins and related polyphenolic compounds [62]. This finding is particularly important because obesity and type 2 diabetes are closely linked, and controlling blood glucose levels is essential for preventing complications associated with these conditions. The study emphasizes the potential of encapsulated Andean berry anthocyanins as a natural strategy to enhance blood glucose regulation and suggests that this technology could be used to develop functional products for individuals at risk of developing type 2 diabetes.”

  1. Reviewer: Current version is overly promotional. Rewrite to reflect the evidence—highlight promising phytochemistry and preclinical results, but stress the absence of strong clinical support. End with a roadmap for roadmap rather than marketing language.

  • Author’s response: The entire document has been revised, and the tone has been adjusted to be more neutral and critical.

We appreciate the significant time and effort the reviewer has dedicated to improving our manuscript. We hope the corrections we have made address the reviewer’s comments.

Prof. Dr. Silvia Andrea Quijano-Pérez

Facultad de Ciencias Básicas

Universidad Santiago de Cali

Corresponding Author 1

Prof. Dr. Sandra Sulay Arango-Varela

Grupo de Investigación Biología Médica

Facultad de Ciencias Exactas y Aplicadas

Instituto Tecnológico Metropolitano.

Corresponding Author 2

Round 2

Reviewer 1 Report

Comments and Suggestions for Authors

The corrections are appropriate and sufficient and improve the quality of the work

Reviewer 2 Report

Comments and Suggestions for Authors

Accept be suggested.

Reviewer 4 Report

Comments and Suggestions for Authors

The revised version of your manuscript shows clear improvement in both structure and scientific content. You have effectively addressed the reviewers’ previous comments, clarified the objectives, and strengthened the discussion of the phytochemical composition and health-promoting potential of Vaccinium meridionale. The presentation is now clearer, and the references are appropriate and up to date.